# Metabolic response to point mutations reveals principles of modulation of *in vivo* enzyme activity and phenotype

Sanchari Bhattacharyya[1,*] (ID), Shimon Bershtein[2], Bharat V Adkar[1] (ID), Jaie Woodard[1] & Eugene I Shakhnovich[1,**] (ID)

## Abstract

**The relationship between sequence variation and phenotype is poorly understood. Here, we use metabolomic analysis to elucidate the molecular mechanism underlying the filamentous phenotype of *E. coli* strains that carry destabilizing mutations in dihydrofolate reductase (DHFR). We find that partial loss of DHFR activity causes reversible filamentation despite SOS response indicative of DNA damage, in contrast to thymineless death (TLD) achieved by complete inhibition of DHFR activity by high concentrations of antibiotic trimethoprim. This phenotype is triggered by a disproportionate drop in intracellular dTTP, which could not be explained by drop in dTMP based on the Michaelis–Menten-like *in vitro* activity curve of thymidylate kinase (Tmk), a downstream enzyme that phosphorylates dTMP to dTDP. Instead, we show that a highly cooperative (Hill coefficient 2.5) *in vivo* activity of Tmk is the cause of suboptimal dTTP levels. dTMP supplementation rescues filamentation and restores *in vivo* Tmk kinetics to Michaelis–Menten. Overall, this study highlights the important role of cellular environment in sculpting enzymatic kinetics with system-level implications for bacterial phenotype.**

**Keywords** filamentation; fractal kinetics; *in vivo* enzyme activity; metabolomics; thymine limitation
**Subject Categories** Metabolism; Microbiology, Virology & Host Pathogen Interaction
**Mol Syst Biol. (2021) 17: e10200**

## Introduction

Understanding genotype–phenotype relationship is a central problem in modern biology. Mutations affect various layers of cellular organization, the mechanistic details of which remain far from being understood. Mutational effects propagate up the ladder of cellular organization from physico-chemical properties of biomolecules up to cellular properties, finally affecting system-level properties, like the epigenome, transcriptome, proteome, metabolome, or the microbiome. Collectively, all layers in this multi-scale genotype–phenotype relationship dictate the fitness/phenotypic outcome of the mutations at the organism level. It has been shown, using the concept of a biophysical fitness landscape, that it is possible to predict fitness effects of mutations from a knowledge of molecular and cellular properties of biomolecules (Dykhuizen *et al*, 1987; Lunzer *et al*, 2005; Dean & Thornton, 2007; Bershtein *et al*, 2015b; Rodrigues *et al*, 2016; Adkar *et al*, 2017; Adkar *et al*, 2019) as well as using system-level properties like proteomics and transcriptomics (Bershtein *et al*, 2015a; Bershtein *et al*, 2017). The metabolome which is represented by the metabolite profile of the cell is a more recent advancement in the -omics technology (Bennett *et al*, 2009). Metabolites represent end products of biochemical pathways; hence, they are downstream to other -omics data and therefore closest to the phenotype. Hence, metabolomics is widely recognized now as an important stepping stone to relate genotype to phenotype (Fiehn, 2002; Patti *et al*, 2012; Bhattacharyya *et al*, 2016; Johnson *et al*, 2016; Zampieri & Sauer, 2017; Bhattacharyya *et al*, 2018; Handakumbura *et al*, 2019; Rodrigues & Shakhnovich, 2019; Harrison *et al*, 2020). In the recent past, high-throughput studies have been dedicated to understanding how genetic variations lead to changes in metabolic profile of the cell (Mulleder *et al*, 2016; Fuhrer *et al*, 2017). Though vast knowledge is available in terms of how mutations perturb metabolite levels either in the local vicinity or distant in the network, a mechanistic knowledge of how such changes modulate phenotypic outcomes is lacking.

In this work, we use targeted metabolomics to understand the mechanistic basis of how destabilizing mutations in the essential core metabolic enzyme of *E. coli* Dihydrofolate Reductase cause pronounced (> 10 times of the normal cells) filamentation of bacteria. DHFR catalyzes conversion of dihydrofolate to tetrahydrofolate, which is an essential one-carbon donor in purine, pyrimidine and amino acid biosynthesis pathways. Though filamentation is a documented effect of trimethoprim (Tmp) (antibiotic that inhibits

1  Department of Chemistry and Chemical Biology, Harvard University, Cambridge, MA, USA
2  Department of Life Sciences, Ben-Gurion University of the Negev, Beer-Sheva, Israel
   *Corresponding author. Tel: +1 617 763 8778; E-mail: sbhattacharyya@fas.harvard.edu
   **Corresponding author. Tel: +1 617 495 4130; E-mail: shakhnovich@chemistry.harvard.edu

bacterial DHFR) treatment, it is mostly associated with thymineless death (TLD) (Sangurdekar *et al*, 2011) due to loss of thymidine nucleotides (dTTP) (Kwon *et al*, 2010) and consequent SOS response (Sangurdekar *et al*, 2010; Sangurdekar *et al*, 2011). In this study, although mutant DHFR strains incur a sharp drop in thymidine nucleotides (dTMP, dTDP, and dTTP) and elicit SOS response, contrary to expectations, we found that filamentation is completely *reversible*. Using a range of Tmp concentrations, we delineated the conditions required for TLD which is complete inhibition of DHFR activity achieved by lethal doses of Tmp, while mutant DHFR strains only incur *partial* loss of activity. Additionally, we found that mutant strains have disproportionately low levels of dTDP (and hence dTTP) primarily due to the strongly cooperative *in vivo* activity of the downstream essential pyrimidine biosynthesis enzyme thymidylate kinase (Tmk), which phosphorylates dTMP to dTDP. This is in stark contrast to its Michaelis–Menten (MM) activity profile observed *in vitro* and is therefore manifested in terms of reduced recovery of intracellular dTDP/dTTP even as dTMP begins to increase. Surprisingly, supplementation of external dTMP in the medium which rescues *in vivo* dTDP levels and filamentation switches the *in vivo* Tmk activity curves to the conventional "*in vitro* like" MM kinetics. The cooperative enzyme activity is best explained by diffusion limitation of substrate dTMP *in vivo*, possibly due to substrate channeling and metabolon formation. This idea is also supported by Hill-like and hyperbolic *in vivo* activity curves of a number of other *E. coli* proteins. Overall, this study highlights the pleiotropic nature of mutations and the way in which the complex cellular environment and metabolic network modulates *in vivo* enzyme activity and organismal fitness.

# Results

## Several chromosomal mutations in folA gene give rise to slow growth and filamentation of *E. coli*

Earlier, we had designed a group of highly destabilizing chromosomal DHFR mutations in *E. coli* MG1655 (W133V, V75H+I155A, I91L+W133V, and V75H+I91L+I155A) that cause very slow growth at 37 and 42°C (Bershtein *et al*, 2012; Bershtein *et al*, 2013) and Fig 1A, see Materials and Methods for details about the strains). While WT, W133V, and V75H+I155A could grow up to 42°C, mutants I91L+W133V and V75H+I91L+I155A could only survive up to 40°C. To understand the effects of these mutations on bacterial morphology, we grew mutant DHFR strains under two different growth conditions: M9 minimal media without and with supplementation with casamino acids (mixtures of all amino acids, except tryptophan, see Materials and Methods). In minimal media, median cell lengths of some mutants (W133V and V75H+I155A at 42°C) were smaller compared to WT, while I91L+W133V (at 40°C) was marginally longer than WT (Fig 1C). However, when M9 minimal medium was supplemented with amino acids, we found that cells carrying these mutations were pronouncedly filamentous. Figure 1B shows live cell DIC images of wild-type (WT) and I91L+W133V mutant DHFR strains at 30, 37, and 42°C (40°C for I91L+W133V strain) (see Fig EV1A for images of other low fitness mutant strains). In parallel to the detrimental effect of temperature on fitness, we noted that the morphologies were also temperature-sensitive. I91L+W133V and

V75H+I91L+I155A strains exhibited a 1.5- to 1.75-fold increase (comparatively to WT) in the average cell length at 37°C (Fig 1D), while W133V and V75H+I155A were not elongated at 37°C. The latter, however, showed an increase up to 2.0- to 2.3-fold over WT cell lengths at 42°C (Fig 1E). Strains I91L+W133V and V75H+I91L+I155A showed 1.8- to 2.0-fold increase in the average cell length at 40°C, with some cells reaching up to 20 μm in length (about 10-fold increase). Besides temperature of growth, since filamentation was also strongly dependent on availability of amino acids in the growth medium, it seemed likely that it was the result of a metabolic response due to partial loss of DHFR function.

## Filamentation is due to drop in DHFR activity

DHFR is a central metabolic enzyme that is involved in conversion of dihydrofolate to tetrahydrofolate, and the latter is an important 1-carbon donor in the biosynthesis of purines, pyrimidines, and certain amino acids like glycine and methionine. Earlier, we had reported that these mutant DHFR strains had very low abundance of the mutant proteins in the cell (Bershtein *et al*, 2013; Bershtein *et al*, 2015a) and Fig EV1B), an effect that could be rescued by deletion of Lon protease or by overexpressing chaperones like GroEL-ES (Bershtein *et al*, 2013). We, therefore, reasoned that filamentation could be a result of drop in DHFR activity in these cells. To confirm this, we supplemented the *E. coli* strains carrying chromosomal DHFR mutations with WT DHFR expressed from a plasmid and found that both filamentation (Fig 1F) and growth defects (Fig 1G) were fully rescued. On the WT background, expression of extra DHFR resulted in some elongation, presumably due to toxicity of DHFR overexpression (Bhattacharyya *et al*, 2016). We also found that plasmid expression of mutant proteins in WT cells did not result in any filamentation (Fig 1H) or growth defects (Fig 1I). This shows that filamentation is not due to toxicity of the mutant DHFR proteins. We also found that treatment of WT cells with trimethoprim, an antibiotic that targets bacterial DHFR (Fig EV1C), also caused filamentation at concentrations near the MIC (1 μg/ml). At higher concentrations of the drug, there is growth arrest, hence no filamentation, leading to a non-monotonic dependence of cell length on Tmp concentration (Fig EV1, panels D and E at 37 and 42°C, respectively).

## Filamentous strains exhibit imbalance between dTTP and other deoxyribonucleotides

Here, we aimed to determine metabolic changes in mutant strains associated with filamentous phenotype. To that end, we carried out metabolomics analysis under conditions of filamentation (in amino acid supplemented M9 medium) as well as under non-filamentation conditions (in minimal medium). As shown in Fig 1, W133V and V75H+I155A mutants behave similarly (in terms of growth rates, extent of filamentation, and ability to grow up to 42°C), while mutants I91L+W133V and V75H+I91L+I155A behaved similarly. Moreover, I91L+W133V had the lowest growth rate among all mutants, which makes it an ideal candidate to study the extreme effect of mutations. Hence, for metabolomics analyses as well as for several later experiments, we chose one representative example from each of these two groups, namely W133V and I91L+W133V, as well as WT treated with 0.5 μg/ml Tmp.

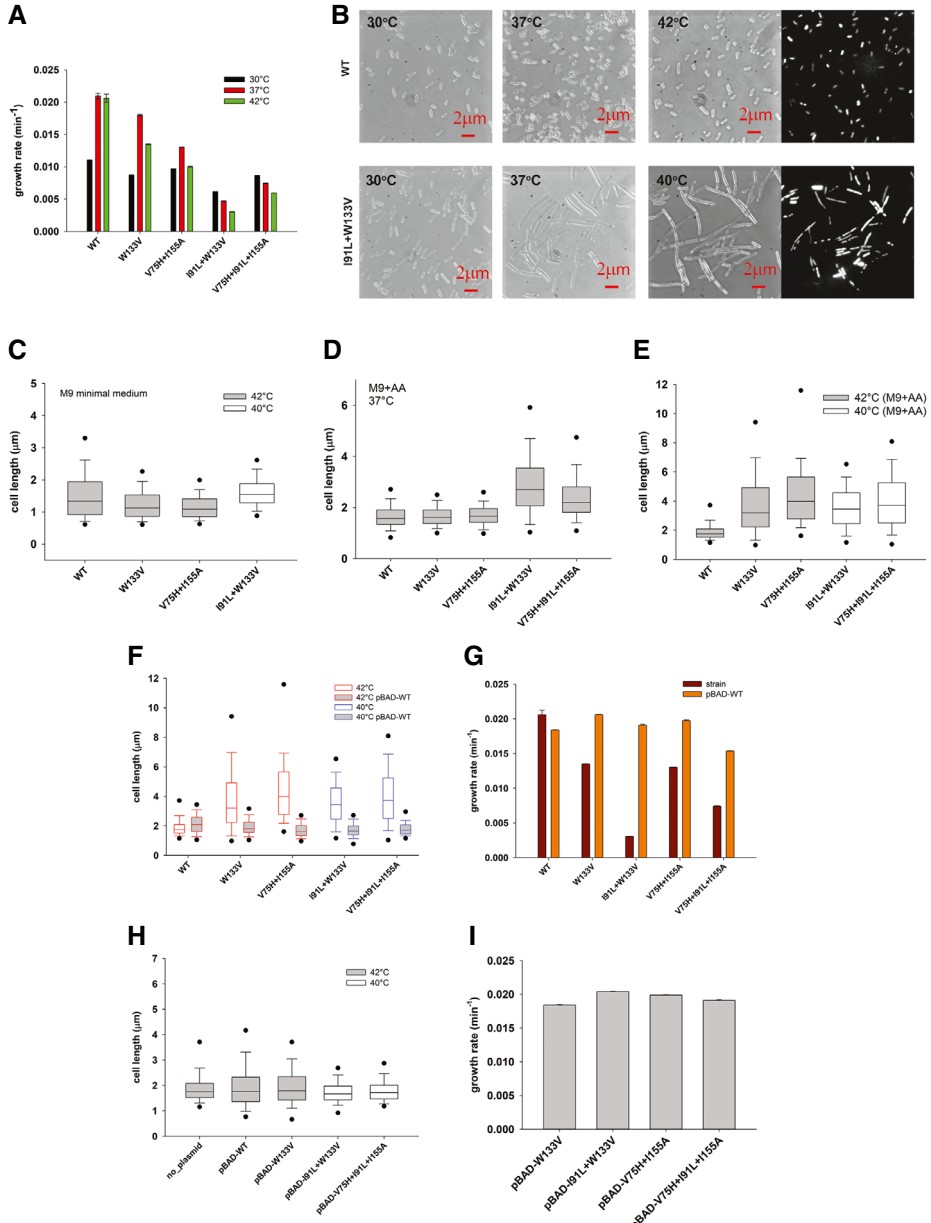

**Figure 1. Destabilizing mutations in DHFR induce filamentous phenotype due to loss of DHFR activity.**

A   Growth rates of mutant DHFR strains at 30, 37, and 42°C. While most mutants grow well at 30°C, they grow very poorly at high temperatures. Error bars represent SEM of three biological replicates.

B–E   Live cell DIC images and DAPI nucleoid staining of WT DHFR and I91L+W133V DHFR strains after being grown at 30, 37, or 42°C (I91L+W133V was grown at 40°C) in M9 medium supplemented with amino acids for 4 h (see Materials and Methods). Cell lengths were measured from the obtained DIC images (see Materials and Methods), and their distribution at 37 and 40 /42°C is shown in (D) and (E) as boxplots (see Materials and Methods). Images of other mutant DHFR strains W133V, V75H+I155A, and V75H+I91L+I155A are presented in related Fig EV1A. (C) Distribution of cell lengths of WT and mutants W133V and V75H+I155A at 42°C (gray box) and I91L+W133V at 40°C (represented as white box) after being grown in M9 minimal medium (without amino acids) for 4 h. Median cell length of W133V and V75H+I155A is significantly smaller than WT (Mann–Whitney test, $P < 0.001$).

F–I   Functional complementation of WT DHFR from an arabinose inducible pBAD plasmid rescues both (F) filamentation and (G) growth defects of mutant strains grown at 42°C (for WT, W133V, and V75H+I155A strains) or at 40°C (for I91L+W133V and V75H+I91L+I155A) in M9 medium supplemented with amino acids. Expression of mutant proteins from pBAD plasmid on the WT background does not result in (H) filamentation or (I) growth defects.

Data information: For the boxplots in panels C, D, E, F, and H, the central band represents the median of the distribution, the box ends represent the 25th and 75th percentiles, and the whiskers represent the 10th and 90th percentiles, while the dots represent the 5th and 95th percentiles. Data were usually obtained from 2-3 biological replicates. The number of cells used to derive the boxplot distributions in the different panels ranges usually between 200 and 450 (please refer to Fig 1 source data for exact number of cells for each dataset). See related Fig EV1B.

Source data are available online for this figure.

We observed that in the absence of amino acids, *when the cells are not filamented,* mutant strains as well as WT cells treated with 0.5 µg/ml Tmp (close to MIC) exhibited very low levels of both purines and pyrimidines (Fig 2A). For example, in strain I91L+W133V, IMP, AMP, and dTMP levels were, respectively, 17, 30, and 5% of WT levels, while dTTP levels were below the detection limit. Methionine and glycine biosynthesis requires tetrahydrofolate derivatives; hence, expectedly, methionine levels were only 1–3% in mutant strains (Fig 2B). Overall, we conclude that large drop in methionine and purine (IMP) levels presumably stalls protein/RNA synthesis. Since increase in cell mass is essential for filamentation, cells under this condition are not filamented.

In the presence of 1% casamino acids, several but not all amino acids showed a marked increase in abundance (Fig 2D). Methionine levels rose to 40% of WT levels for I91L+W133V mutant, while aspartate/asparagine, glutamine, histidine, and tryptophan levels also showed a significant increase. Particularly interesting was the fact that purine levels were substantially rescued upon addition of amino acids (Fig 2C). IMP showed the maximal effect, increasing 10- to 15-fold over its levels in the absence of amino acids. ATP, ADP, AMP, and GMP also showed similar trends. Since the product of DHFR is eventually used in the synthesis of methionine, IMP, and dTMP (Fig EV2A), we hypothesize that addition of methionine in the medium allows higher amounts of 5,10-methylene-THF to be channeled toward synthesis of purines and pyrimidines. Moreover, both *de novo* purine and pyrimidine biosynthesis pathways require aspartate and glutamine (Fig EV2B and C), which were otherwise low in minimal medium. Overall, the metabolomics data suggest that in the presence of added amino acids, protein and RNA synthesis are no longer stalled, and therefore growth, which is prerequisite for filamentation, can happen.

Though dTMP levels increased to about ~10% of WT levels in I91L+W133V and ~50% in W133V, surprisingly, dTTP levels (thymidine derivative that is incorporated in the DNA) were only about 1% of WT levels in I91L+W133V and 18% for W133V (Fig 2C). In contrast, dATP and dCTP levels were very high (Fig 2C). We hypothesize that as cellular growth continues, misbalance in the concentrations of deoxynucleotides may lead to erroneous DNA replication, induction of SOS response, and blocked cell division. Indeed, in our previous study, proteomics and transcriptomics analyses showed that several SOS response genes were upregulated in I91L+W133V and V75H+I91L+I155A strains at 37°C (Bershtein et al, 2015a). We note that the imbalance of deoxynucleotides due to extremely low dTTP levels happens even in the absence of added amino acids (Fig 2A); however, this may not cause erroneous DNA replication as the cell cannot duplicate its cellular mass due to depletion of methionine and purines (so-called stringent response), which are essential for protein and RNA synthesis, and hence does not attempt cell division. Therefore, this condition does not result in filamentation.

### Filamentation and SOS response: deletion of recA rescues filamentation

We quantified the expression of several SOS genes *recA, recN,* and *sulA* under different supplementation conditions for mutant DHFR strains as well as Tmp-treated WT cells. Mutants grown at low temperature and those grown in the absence of amino acids were not elongated, and not surprisingly, they did not elicit any SOS response (Fig 3A–C). In comparison, all conditions that are associated with filamentation (I91L+W133V mutant at 37°C, W133V and I91L+W133V at 42°C) induced strong SOS response (Fig 3A–C). The levels of induction of these genes were greatly reduced in the presence of dTMP, consistent with lack of filamentation under this condition (Fig 3A–C). Similar trends were also observed for WT treated with Tmp.

Overexpression of RecA is the key trigger of the SOS response to DNA damage. RecA cleaves the dimeric LexA repressor to turn on the genes that are under the SOS box (e.g., sulA and uvr proteins). SulA inhibits the cell division protein FtsZ, eventually causing filamentation. To understand the role of *recA* and *sulA* in our study, we treated ∆*recA* and ∆*sulA* strains with near-MIC levels of the antibiotic Tmp. As expected, ∆*recA* strain did not show filamentation upon Tmp treatment (Fig 3D), clearly highlighting its definitive role in filamentation. However, ∆*sulA* strain continued to filament (Fig 3 D), suggesting possible role of sulA-independent pathways. These data, however, do not negate out the role of sulA, since sulA is highly upregulated in mutant strains and is a well-known inhibitor of the cell division protein FtsZ. A poorly characterized segment of the *E. coli* genome called the e14 prophage that harbors the *sfiC* gene has been implicated to play a role in SOS-dependent but sulA-independent filamentation (Maguin *et al*, 1986). To find out if this e14 prophage region and the sfiC gene are involved in a sulA-independent SOS pathway in our case, we used a mutant *E. coli* strain that is knocked out for the e14 prophage region and found that this strain does not filament upon Tmp treatment (Fig 3D), indicating that *sfiC* might be one of the players involved in elongation, though the exact mechanism of action through this pathway is unclear. Regardless of the mechanism, these results clearly establish the role of the SOS pathway toward filamentation in mutant DHFR strains, which in turn is due to imbalance between dTTP and other deoxynucleotides in the cell.

### Filamentation in mutant DHFR strains is not due to TLD and is reversible

A moderate level of thymine deficiency—thymine limitation—affects cell shape by increasing its diameter but not the length, making cells less rod-like, and certainly not filamentous (Pritchard & Zaritsky, 1970; Zaritsky & Pritchard, 1973). A more severe level of thymine deprivation causes, under certain conditions, thymineless death (TLD), a phenomenon accompanied by cell filamentation (Bazill, 1967; Ahmad *et al*, 1998; Zaritsky *et al*, 2006; Sangurdekar *et al*, 2011). Prior work (Sangurdekar *et al*, 2010; Sangurdekar *et al*, 2011) also showed that thymine limitation is metabolically different from thymineless death, as DNA damage and SOS induction happens only in the latter, and cell death happening due to erroneous DNA replication. The phenotypic, metabolic as well as expression signatures of mutant DHFR strains seemed, therefore, to resemble a case of thymine deprivation and TLD. To find out if mutant strains represented a case of TLD, we assessed the viability of mutant strains at 30°C (permissive temperature) on solid media after several hours of growth at 42°C (restrictive temperature) in the presence of amino acids (filamentation condition). Sangurdekar *et al* (2010) reported exponential loss in viability in TLD after one hour of growth under thymineless conditions. The idea was that if

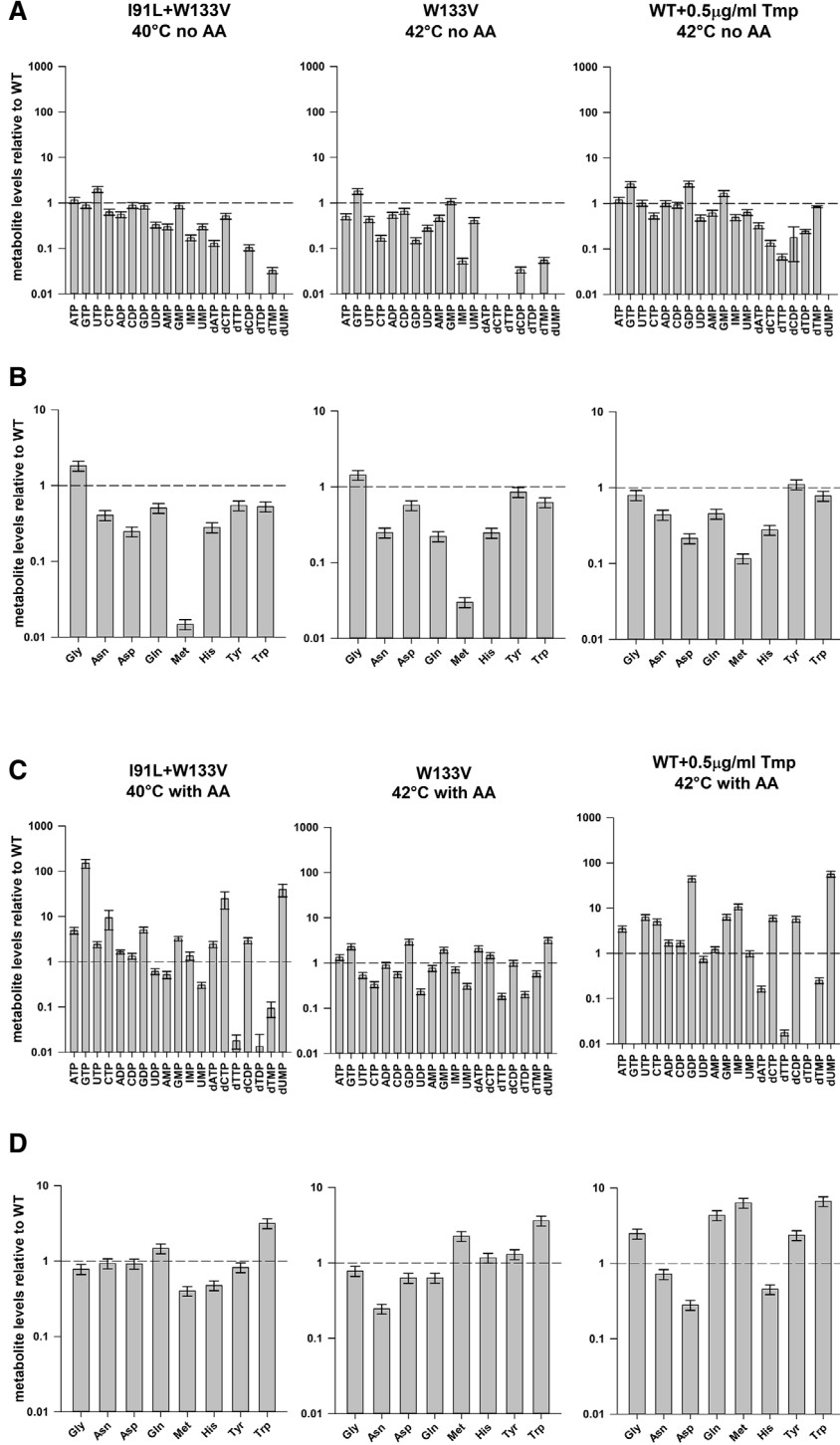

**Figure 2.  Metabolomics of mutant DHFR strains in minimal media without or with added amino acids.**

A–D  Panels (A) and (B) show abundance of selected nucleotides and amino acids for mutants I91L+W133V and W133V as well as WT strain treated with 0.5 µg/ml of Tmp (trimethoprim) after 4 h of growth at 40/42°C in M9 minimal medium (condition of no filamentation), while (C) and (D) represent nucleotide and amino acid abundances after 4 h of growth in amino acid supplemented M9 medium at 40/42°C (condition of filamentation). Concentration of all metabolites was normalized to WT levels at 4 h when grown under similar conditions. In minimal medium (B), methionine levels are extremely low, which recover largely in panel (D). Levels of purines (IMP, AMP) are also largely rescued with amino acid supplementation; however, dTMP, dTDP, and dTTP levels remain extremely low. Error bars represent SEM of at least three biological replicates (see Materials and Methods). The dashed line in each plot represents value of 1 for WT.

Source data are available online for this figure.

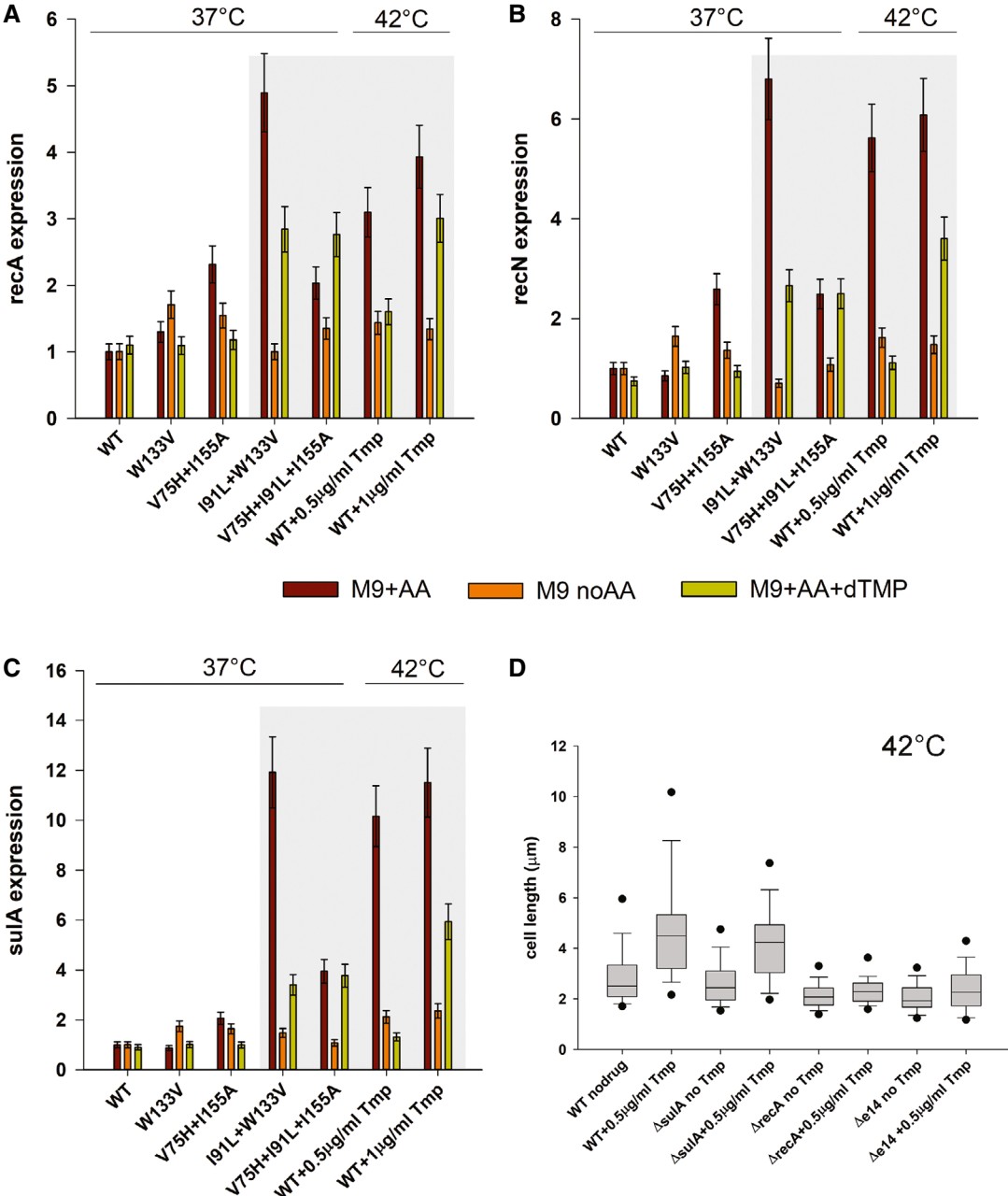

**Figure 3. Filamentation in mutant DHFR strains is associated with strong SOS response.**

A–C  Expression of (A) *recA* (B) *recN*, and (C) *sulA* genes measured by quantitative PCR when WT and mutant strains are grown in M9 medium with or without supplementation of amino acids or dTMP. WT and mutant strains were grown for 4 h of growth in the indicated medium at 37°C, while WT treated with different concentrations of Tmp were grown for 4 h at 42°C. Brown bars (M9+AA) in the gray shaded area correspond to filamentation conditions and these are associated with pronounced upregulation of all three SOS genes. On the other hand, conditions with loss of filamentation (with dTMP or no supplementation) show much less expression. Error bars represent SD of 2–3 biological replicates (see Materials and Methods).

D  Treatment of WT *E. coli* cells with sub-MIC concentration of Tmp (0.5 µg/ml) leads to filamentation at 42°C when grown in amino acid supplemented medium. However, a *recA* knock-out strain under similar condition shows no elongation, indicating the role of SOS pathway in filamentation. A *sulA* knock-out continues to elongate, indicating the role of *sulA*-independent pathways. An *E. coli* strain deleted for the e14 prophage region however showed no filamentation upon Tmp treatment, indicating that sfiC gene in the e14 region might be one such sulA-independent player. The central band in the boxplots represents the median of the distribution, the box ends represent the 25th and 75th percentiles, and the whiskers represent the 10th and 90th percentiles, while the dots represent the 5th and 95th percentiles. Data were usually obtained from 2-3 biological replicates. The number of cells used to derive the boxplot distributions in the different panels ranges usually between 100 and 200 (please refer to Fig 3 source data for exact number of cells for each dataset).

Source data are available online for this figure.

cells underwent TLD under conditions of filamentation, they would no longer be able to resume growth/form colonies at permissive temperature. To that end, we induced filamentous phenotype by incubating W133V mutant cells for 4 h at 42°C and then monitored cell recovery at room temperature (see Materials and Methods). Figure 4A shows a representative example of morphology of W133V

filament that begins to undergo slow division initiated at the poles at permissive temperature. Most progeny cells of normal size appeared after 5–6 h of growth at low temperature, indicating no loss in viability.

Earlier studies have also shown that inhibition of DHFR activity by trimethoprim (Tmp) under conditions of both amino acids and

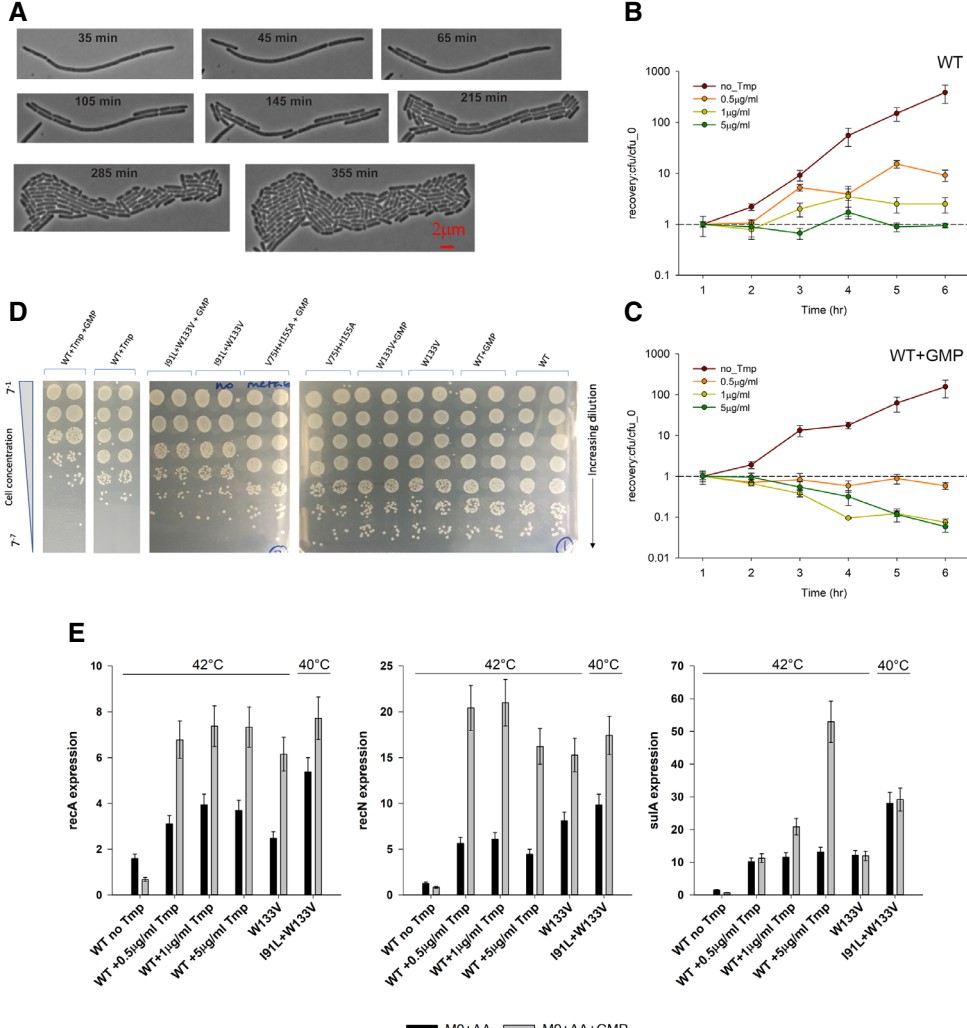

**Figure 4. Filamentation in mutant DHFR strains is completely reversible and does not represent TLD.**

A  Mutant W133V was grown in amino acid supplemented M9 medium (M9+AA) for 4 h at 42°C and subsequently placed on M9 agar pads, and their growth was monitored at room temperature. Shown are phase contrast images taken from different time points throughout the time-lapse experiment. Unlike cells experiencing TLD, an irreversible phenomenon, W133V DHFR cells recover and resume growth at low temperature.

B, C  WT cells were treated with different concentrations of Tmp at 42°C for varying amounts of time in amino acid supplemented M9 medium (panel B) or in M9 media supplemented with both amino acids and GMP (panel C), following which they were spotted on M9+AA plates and allowed to grow at 30°C. Colonies were counted next day. In the presence of only amino acids, there was no loss in viability for any concentration of Tmp (panel B), despite extensive filamentation (Fig EV1E). In contrast, in the presence of amino acids and GMP, the cells showed sharp loss in viability when grown at high Tmp concentrations. In both panels, error bars represent SD of two biological replicates. The dashed line represents cfu at one-hour timepoint.

D  WT (with and without 5 µg/ml Tmp) and mutants were grown for 6 h at 42°C in M9+AA medium without or with GMP and subsequently diluted serially and spotted on M9+AA agar plates and allowed to grow at 30°C till visible colonies were formed. While WT treated with high Tmp shows loss in cfu, indicating TLD, no loss in viability was observed for WT or mutants.

E  Expression of SOS response genes *recA*, *recN*, and *sulA* when mutant strains and Tmp-treated WT cells are grown in M9 medium supplemented with amino acids (black bars) or in the presence of both amino acids and GMP (gray bars) at 42°C (I91L+W133V was grown at 40°C). Error bars represent SD of 2–3 biological replicates.

purine supplementation leads to TLD (Kwon *et al*, 2010). We found that when WT cells were grown on media supplemented with *both* amino acids and a purine source (GMP), they showed loss of colony-forming units (indicating death) only when subjected to *very high* Tmp concentrations (Fig 4C), similar to Kwon *et al* (2010). On the other hand, supplementation with only amino acids was bacteriostatic under all conditions of Tmp (Fig 4B). Since mutant DHFR strains incur only *partial* loss of DHFR activity, they resemble lower concentrations of Tmp treatment and therefore did not show any substantial loss in colony-forming units even when grown in the presence of amino acids and GMP (Fig 4D), though the extent of SOS response increased substantially under this condition, indicating greater DNA damage (Fig 4E). Despite this, the fact that the mutant DHFR strains retained viability reveals that TLD and cell survival depend crucially on the extent to which DHFR activity is compromised, a condition that is only achieved with lethal doses of Tmp. However, the data also show expression of SOS genes alone cannot differentiate between the bacteriostatic and bactericidal regimes (Fig 4E). Collectively, these experiments disentangle the relationship between the regimes of DHFR activity and filamentation and TLD. This study also highlights the considerable overlap that exists between characteristics of thymine limitation and thymineless death and emphasizes the fact that these two conditions are not an all or none phenomenon.

## Supplementation of dTMP in the medium restores dTDP/dTTP levels and rescues filamentation

Since mutant strains had very low dTMP/dTTP levels, we grew WT and mutant strains in minimal medium that was supplemented with both amino acids and 1 mM dTMP and carried out measurement of cell length as well as metabolomics analysis. Addition of dTMP largely rescued filamentation of mutant strains (Fig 5A). Metabolomics analyses showed that under this condition, I91L+W133V mutant had much higher levels of both dTDP and dTTP relative to WT (Fig 5B). Concentration of the other deoxyribonucleotides dATP/dCTP levels, however, remained high despite addition of dTMP (Fig 5B). Therefore, we conclude that higher amounts of dTTP presumably reduce the imbalance in relative concentrations of deoxynucleotides, thus relieving DNA damage and filamentation (see above).

Moreover, supplementation of dTMP and amino acids also allowed mutants W133V, I91L+W133V, and V75H+I91L+I155A to form higher counts of colony-forming units (cfu) than with only amino acids at their respective filamentation temperatures (Fig 5C shows images for V75H+I91L+I155A at 40°C). However, we note that cfu count for mutants in the presence of dTMP was still orders of magnitude lower than those of WT, indicating that thymine only rescues cell length, and not growth defects. This was also supported by the absence of growth rate rescue with dTMP (Fig 5D).

## Low dTDP/dTMP ratio: possibility of inhibition of thymidylate kinase?

Interestingly, we found that while dTMP levels are low in mutant DHFR strains (10 and 50% of WT levels in I91L+W133V and W133V mutants, respectively), dTDP levels were far lower (1 and 20%) (Fig 2C). On the other hand, the ratios of dTDP to dTTP were approximately equivalent in both WT and mutants.

Supplementation of dTMP in the medium, however, restores the relative abundances of dTMP to dTDP to dTTP to approximately WT level in I91L+W133V (Fig 5B). It raises the possibility that the pyrimidine biosynthesis pathway enzyme thymidylate kinase (Tmk), which phosphorylates dTMP to dTDP, might be inhibited in mutant DHFR strains. Previous reports suggest that dUMP and dCTP can act as competitive inhibitors for Tmk (Nelson & Carter, 1969), and interestingly, we found that both dUMP and dCTP levels were highly upregulated in mutant cells (Fig 2C), due to inefficient conversion of dUMP to dTMP when DHFR activity was reduced (Fig EV2C). To find out if intracellular accumulation of dUMP and dCTP in mutant cells is sufficient to inhibit Tmk, we overexpressed and purified his-tagged Tmk from *E. coli* cells and tested the potential inhibitory effect of dUMP and dCTP *in vitro*. In comparison with its cognate substrate dTMP for which the $K_M$ is 13 μM (Fig 6A), the $K_M$ for dUMP is 450 μM (Fig EV3A), which indicates that dUMP is a much weaker substrate as compared to dTMP. The apparent $K_I$ of dUMP for Tmk was 3.9 mM (Fig EV3B). Given that dUMP concentration inside WT *E. coli* cells is 0.01 mM (Møllgaard & Neuhard, 1983), its concentration inside mutant DHFR cells would be in the range of 0.5 mM (50-fold higher levels), which is not enough to cause substantial inhibition of Tmk *in vivo*. We next carried out an activity assay of Tmk in the presence of varying amounts of dCTP. Depending on magnesium concentration, the $K_I$ of dCTP ranged between 2 and 4 mM (Fig EV3C). Considering intracellular dCTP concentration in WT cells to be 35 μM (Bennett *et al*, 2009), those in mutant cells are about 0.7 mM (20-fold higher levels). Hence, like dUMP, intracellular dCTP levels are too low to show any substantial inhibition of Tmk activity *in vivo*. To conclude, though dUMP and dCTP have the potential to inhibit Tmk activity, their concentrations inside mutant cells are not high enough to achieve that.

## Cooperative enzymatic activity of Tmk *in vivo* explains low dTDP levels

As discussed previously, our metabolomics data show that in mutant cells, dTDP levels fall far more precipitously than dTMP concentrations. If Tmk follows the same Michaelis–Menten-(MM) like dependence on intracellular dTMP concentrations as seen *in vitro*, we find that dTDP levels would never drop as low as the experimentally observed value for mutants (Fig 6B, black line for I91L+W133V) for any assumed intracellular dTMP concentration for WT. To resolve this inconsistency, we attempted to elucidate the *in vivo* activity curve of Tmk enzyme from dTDP and dTMP levels measured from the metabolomics data for WT and mutant cells. To get a broad range of data, we measured metabolite levels for WT and several mutants at different time points during growth both in the absence and presence of different concentrations of external dTMP in the medium. Surprisingly, we found that the data points from our metabolomics experiments traced two different curves depending on whether there was external dTMP in the medium. Data points derived in the *absence* of external dTMP had a long lag, followed by a more cooperative increase (red points in Fig 6C), while the data points corresponding to added dTMP appeared to follow the traditional MM curve (black data points in Fig 6C) and were similar to the enzyme activity of Tmk observed *in vitro*. We fitted both datasets to a Hill equation. For the conditions without added dTMP (red data points of Fig 6C), Hill coefficient of 2.5 was

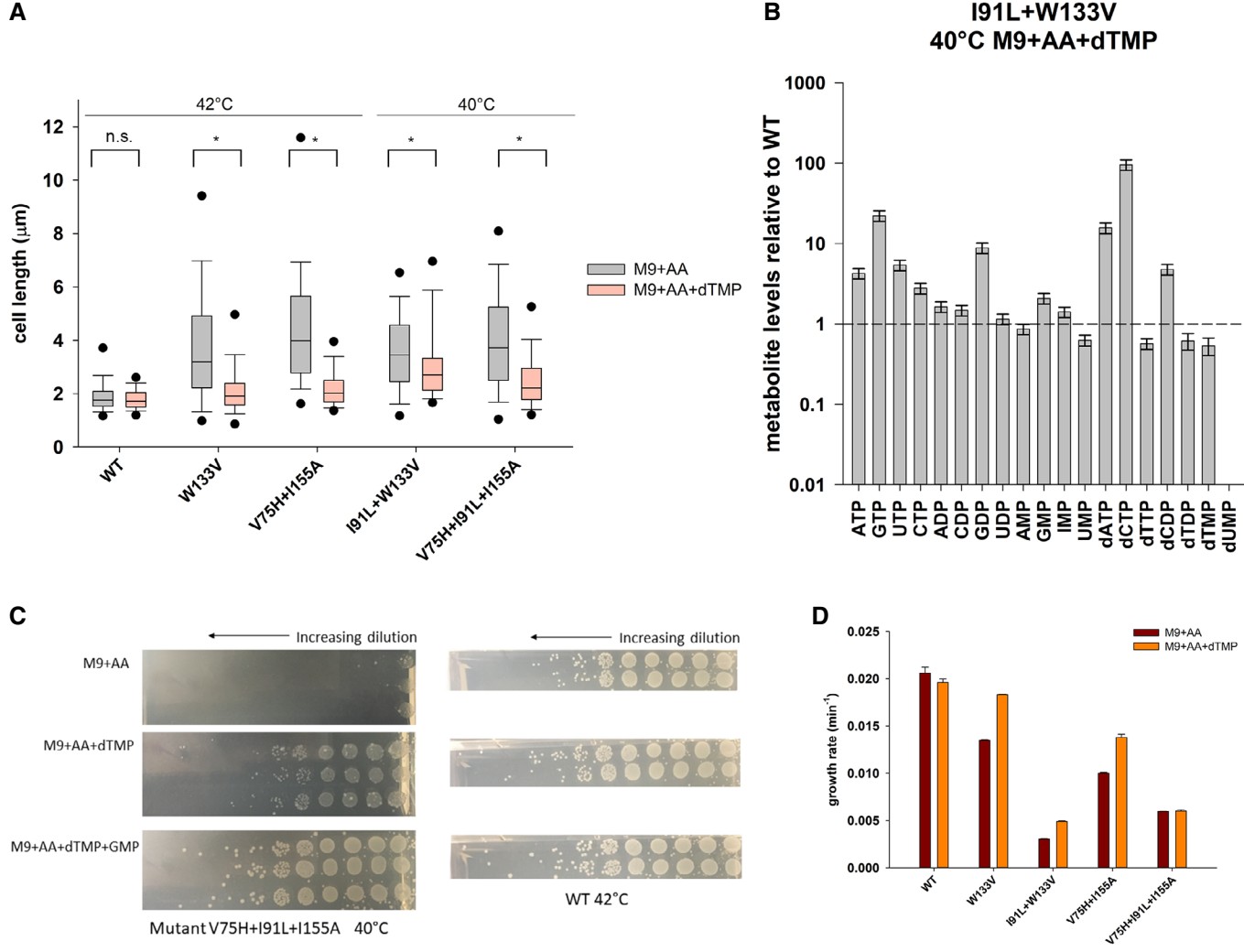

**Figure 5. Supplementation of dTMP alleviates dTDP/dTTP levels and rescues filamentation.**

A   Distribution of cell length of WT and mutant strains when grown in M9 medium supplemented with amino acids (gray) or with both amino acids and 1 mM dTMP (pink). WT, W133V, and V75H+I155A strains were grown at 42°C, while I91L+W133V and V75H+I91L+I155A mutants were grown at 40°C. 1 mM dTMP largely rescues filamentation of mutant strains (*indicates the median cell lengths were significantly different, Mann–Whitney test, $P < 0.001$). The central band in the boxplots represents the median of the distribution, the box ends represent the 25th and 75th percentiles, and the whiskers represent the 10th and 90th percentiles, while the dots represent the 5th and 95th percentiles. Data were usually obtained from 2 to 3 biological replicates. The number of cells used to derive the boxplot distributions in the different panels ranges usually between 150 and 300 (please refer to Fig 5 source data for exact number of cells for each dataset).

B   Abundances of selected nucleotides in I91L+W133V mutant when grown for 4 h at 40°C in M9 medium supplemented with both amino acids and 1 mM dTMP. Metabolite levels were normalized to those of WT grown under similar conditions. dTDP and dTTP levels recover and are now comparable to dTMP levels. Error bars represent SEM of 3 biological replicates. The dashed line represents value of 1 for WT.

C   Mutant V75H+ I91L+I155A grows very poorly (in terms of colony-forming units, cfu) on a minimal media agar plate supplemented with amino acids (M9+AA) at 40°C. Supplementation of additional dTMP increases the cfu by several orders at the same temperature, while supplementation with both pyrimidine (dTMP) and purine (GMP) allows it to grow as good as WT. In comparison, WT was grown at 42°C under different supplementation conditions. In all cases, cultures were 7-fold serially diluted for the next spot. The three rows (two rows for WT) in each condition represent biological replicates.

D   Comparison of growth rates of WT and mutant DHFR strains at 42°C (40°C for I91L+W133V and V75H+I91L+I155A) in minimal medium that is supplemented with amino acids and/or 1 mM dTMP. Except for W133V and to a lesser extent for V75H+I155A, the effect of dTMP on growth rates is only modest. Error bars represent SEM of 3 biological replicates.

Source data are available online for this figure.

obtained suggesting strong positive cooperativity. On the other hand, when dTMP was added, the dTDP versus dTMP curve (black data points in Fig 6C) was fitted with a Hill coefficient of 1.2. However, the fit was not significantly different from the traditional

MM model ($P = 0.36$). Based on the Hill-like curve in Fig 6C, we can say that for intracellular dTMP concentrations below 10 μM, a 10% drop in dTMP levels (as seen for I91L+W133V) would cause dTDP levels to drop to 1% or less (Fig 6B, red line), largely because

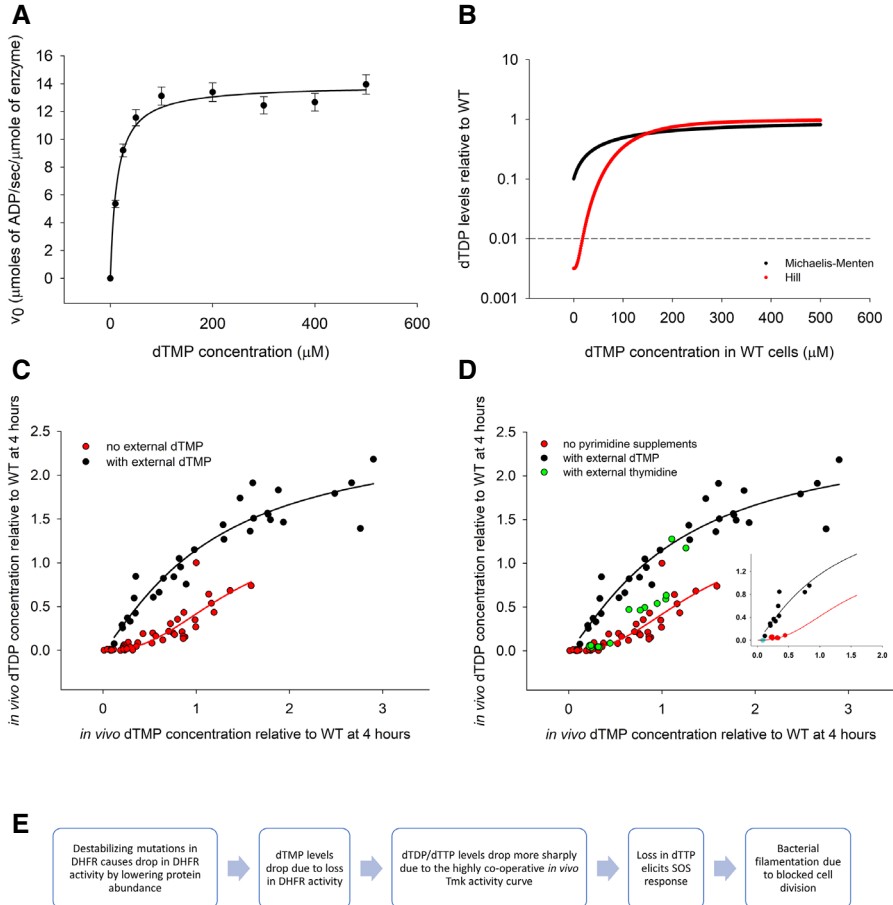

**Figure 6. In vivo enzyme kinetics of Tmk is highly cooperative and different from that in vitro.**

A   *In vitro* activity of purified thymidylate kinase (Tmk) as a function of dTMP concentration shows Michaelis–Menten (MM)-like kinetics. ATP concentration is saturating at 1 mM. The $K_M$ of dTMP is 13 μM. The error bars represent SD of three technical replicates.

B   For a 10-fold drop in intracellular dTMP concentration in mutant relative to WT (as seen in I91L+W133V mutant), we calculate dTDP levels in mutant (relative to WT) as a function of various assumed intracellular concentrations of dTMP in WT (shown along x-axis), as absolute value of this is not known experimentally, assuming MM kinetics (black line) as shown in panel (A) or Hill kinetics with coefficient of 2.5 (red line) as shown in panel (C). The dotted line corresponds to the experimentally observed dTDP ratio of 0.01 for I91L+W133V mutant. This ratio is realized only for the Hill-like curve.

C   Apparent *in vivo* activity kinetics of Tmk enzyme using steady-state dTMP and dTDP levels obtained from metabolomics. The plot includes data from WT, mutants W133V and I91L+W133V, and WT treated with 0.5 μg/ml trimethoprim, obtained at different time points during growth. Data points represent metabolite levels for all individual biological replicates without averaging. The black data points were acquired during growth in the presence of different concentrations of external dTMP (0.25, 0.5, 1, 2, and 5 mM), while red points were from conditions with no external dTMP. Both black and red solid lines represent fit to Hill equation.

D   The green points were acquired during growth of WT and W133V and I91L+W133V mutants in the presence of different concentrations of thymidine, which follow the red curve. The inset plot shows the same graph with selected data points for I91L+W133V mutant. The cyan circle shows I91L+W133V mutant in the absence of any metabolite supplementation, while red and black points indicate metabolite levels following thymidine and dTMP supplementation, respectively.

E   A flow chart summary of the actual chain of events triggered by destabilizing mutations in DHFR that eventually lead to bacterial filamentation, as revealed in this study.

Source data are available online for this figure.

of the long lag. Therefore, a Hill-like dependence of Tmk activity on dTMP concentrations can explain the disproportionately low dTDP levels in mutant strains.

However, we note that metabolomics does not directly report on the kinetics of an enzyme *in vivo*. Rather, it reports on the steady-state levels of metabolites present inside the cell at any given time. Moreover, unlike *in vitro* activity measurement conditions where an enzyme functions in isolation, *in vivo* Tmk is a part of the pyrimidine biosynthesis pathway that involves several sequentially acting enzymes as follows:

$$E_1 + S_1 \underset{k_{-1}}{\overset{k_1}{\rightleftharpoons}} E_1 S_1 \underset{k_{-2}}{\overset{k_2}{\rightleftharpoons}} E_1 + S_2$$
$$E_2 + S_2 \underset{k_{-3}}{\overset{k_3}{\rightleftharpoons}} E_2 S_2 \underset{k_{-4}}{\overset{k_4}{\rightleftharpoons}} E_2 + S_3 \rightleftharpoons .....$$

(1)

where all metabolites ($S_1$, $S_2$, $S_3$...) are at steady state.

To shed light on whether steady-state metabolomics data can give us insight into the kinetics of enzymes *in vivo*, we carry out a simple exercise as follows: For a system (like equation 1) where a series of enzymes work sequentially, we assume two scenarios.

Case 1: We assume that each enzyme in this pathway follows Michaelis–Menten kinetics and that the *product* of each enzyme has very low affinity to bind the enzyme back. Using the detailed derivation in Materials and Methods, we arrive at the following equation:

$$[S_2] = \frac{k_2 K_{M2} [E_1]_0 [S_1]}{k_4 K_{M1} [E_2]_0 + \left(k_4 [E_2]_0 - k_2 [E_1]_0\right)[S_1]} = \frac{A[S_1]}{B + C[S_1]} \quad (2)$$

where $A = k_2 K_{M2} [E_1]_0$, $B = k_4 K_{M1} [E_2]_0$, $C = \left(k_4 [E_2]_0 - k_2 [E_1]_0\right)$ and $K_{M1}$ and $K_{M2}$ are the Michaelis constants of enzymes $E_1$ and $E_2$ for $S1$ and $S2$, and $[E_i]_0$ is the total concentration (i.e., free+bound) of an *i-th* enzyme. Equation (2) tells us that if individual enzymes follow MM kinetics (assumptions of Case 1), then we expect that the steady-state concentration of any product in the pathway will also follow a hyperbolic or MM-like dependence on its substrate concentration.

Case 2: We assume that $E_1$ or $E_2$ or both follow Hill-like kinetics (due to reasons, we elaborate in the Discussion) in the following form:

Initial rate $\nu_0 = \dfrac{\nu_{\max} [S]^m}{K_M + [S]^m}$, where "$m$" is the Hill coefficient,

then for the pathway of enzymes at steady state (as in Equation 1), $S2$ has the following dependence on $S1$:

$$[S_2]^n = \frac{k_2 K_{M2} [E_1]_0 [S_1]^m}{k_4 K_{M1} [E_2]_0 + \left(k_4 [E_2]_0 - k_2 [E_1]_0\right)[S_1]^m} = \frac{A[S_1]^m}{B + C[S_1]^m}, \quad \text{where}$$

"$m$" and "$n$" are the Hill coefficients of consecutive enzymes $E_1$ and $E_2$.

Hence,

$$[S_2] = \left[\frac{A[S_1]^m}{B + C[S_1]^m}\right]^{1/n} \quad (3)$$

As shown in Materials and Methods and Fig EV6, equation (3) tells us in case 2 scenario, $S2$ will show a Hill-like dependence on $S1$ with positive cooperativity with the assumption that $m > n$.

The above analyses with both cases 1 and 2 suggest that the dependence of the steady-state concentrations of metabolites on each other along a metabolic pathway is reflective of the kinetics of the concerned enzyme *in vivo*. In other words, since the Tmk activity curve *in vivo* is Hill-like in the *absence* of external dTMP, and MM-like in the presence of added dTMP (Fig 6C), it is reasonable to infer that *kinetics* of Tmk *in vivo* is Hill-like without added dTMP and MM-like in the presence of external dTMP. Of course, this analysis assumes that ATP, the second substrate of Tmk, is not limiting and therefore kinetics of Tmk is pseudo-unimolecular with respect to dTMP. Indeed, our metabolomics data validate the assumption of pseudo-unimolecular *in vivo* kinetics of Tmk by showing that even in mutant DHFR strains, ATP is not depleted, hence not limiting.

### Supplementation of thymidine retains cooperative behavior of Tmk *in vivo*

dTMP, the substrate of Tmk, comes from two different sources inside the cell: the *de novo* pyrimidine biosynthesis pathway through conversion of dUMP to dTMP by thymidylate synthase and the pyrimidine salvage pathway through conversion of thymine to thymidine to dTMP. Mutant DHFR strains which are unable to efficiently convert dUMP to dTMP through the *de novo* pathway due to

reduced folate activity rely substantially on the salvage pathway for their dTMP supply, as has been shown for catalytically inactive mutants of DHFR (Rodrigues & Shakhnovich, 2019). Hence, we next asked the question: What happens to the Tmk activity curve *in vivo* if dTMP is produced (largely) through the salvage pathway instead of being directly supplied from an external source? To that end, we supplemented the growth medium with intermediates from the salvage pathway, namely thymine and thymidine. While supplementation of thymidine increased intracellular dTMP levels for WT as well as mutants (Fig EV4A), the dTDP versus dTMP levels followed the Hill-like curve (Fig 6D, inset figure shows data for the I91L+W133V mutant in the absence of supplementation and in the presence of added thymidine and dTMP). This shows that direct supplementation of the substrate (dTMP) of Tmk results in very different enzyme kinetics compared to when a precursor of dTMP is supplied externally.

Quite surprisingly and contrary to WT, *mutant* strains did not use external thymine base toward increasing intracellular dTMP (Fig EV4A), though it was up taken by the cells (Fig EV4B). We ruled out inhibition of DeoA enzyme (which interconverts thymine and thymidine) in mutants, as thymidine supplementation increases thymine levels significantly (Fig EV4C). However, I91L+W133V mutant had considerably lower level of deoxy-D-ribose-1-phosphate (dR-1P) (Fig EV4D) which is used as a substrate by the enzyme DeoA to synthesize thymidine from thymine. This strain also accumulated large excess of deoxy-D-ribose-5-phosphate (dR-5P) (Fig EV4D), indicating that isomerization of the sugar dR-1P to dR-5P through DeoB enzyme might be one of the reasons for the lack of thymine utilization. This scenario is supported by the recent finding that cells that evolved to grow on small amounts of thymine supplement on the background of inactive DHFR mutationally deactivated DeoB thus blocking the channeling of dR-5P toward glycolysis and providing sufficient amount of dR-1P toward thymidine synthesis in the salvage pathway (Rodrigues & Shakhnovich, 2019).

### Is cooperativity a general signature of enzyme activity *in vivo*?

Since *in vivo* kinetics of thymidylate kinase is cooperative and Hill-like, we next sought to find out if this is a general characteristic of metabolic enzymes *in vivo*. To that end, we looked at the enzyme adenylate kinase (AK), which is an essential enzyme in *E. coli* that interconverts adenylate currencies AMP, ADP, and ATP. Importantly, from a previous study (Adkar *et al*, 2019), we had metabolomics data available for several stabilizing and destabilizing mutants of AK in the absence and presence of external AMP. This provided us a handle to span a large range of intracellular AMP and ADP concentrations, in turn allowing us to get an insight into the kinetics of AK enzyme *in vivo*. When we overlay the relative AMP and ADP concentrations for all mutants (Fig 7A), we find that much unlike Tmk, the *in vivo* activity curves of AK obey Michaelis–Menten-like kinetics both in the presence and absence of external AMP supplementation.

Intrigued by the AK data, we set out to look for similar data for other *E. coli* enzymes. However, in the absence of direct mutational data for any other enzyme, it is difficult to obtain a large enough dynamic range of intracellular metabolite concentrations. In case of Tmk, it was possible to obtain large variations in dTMP levels as

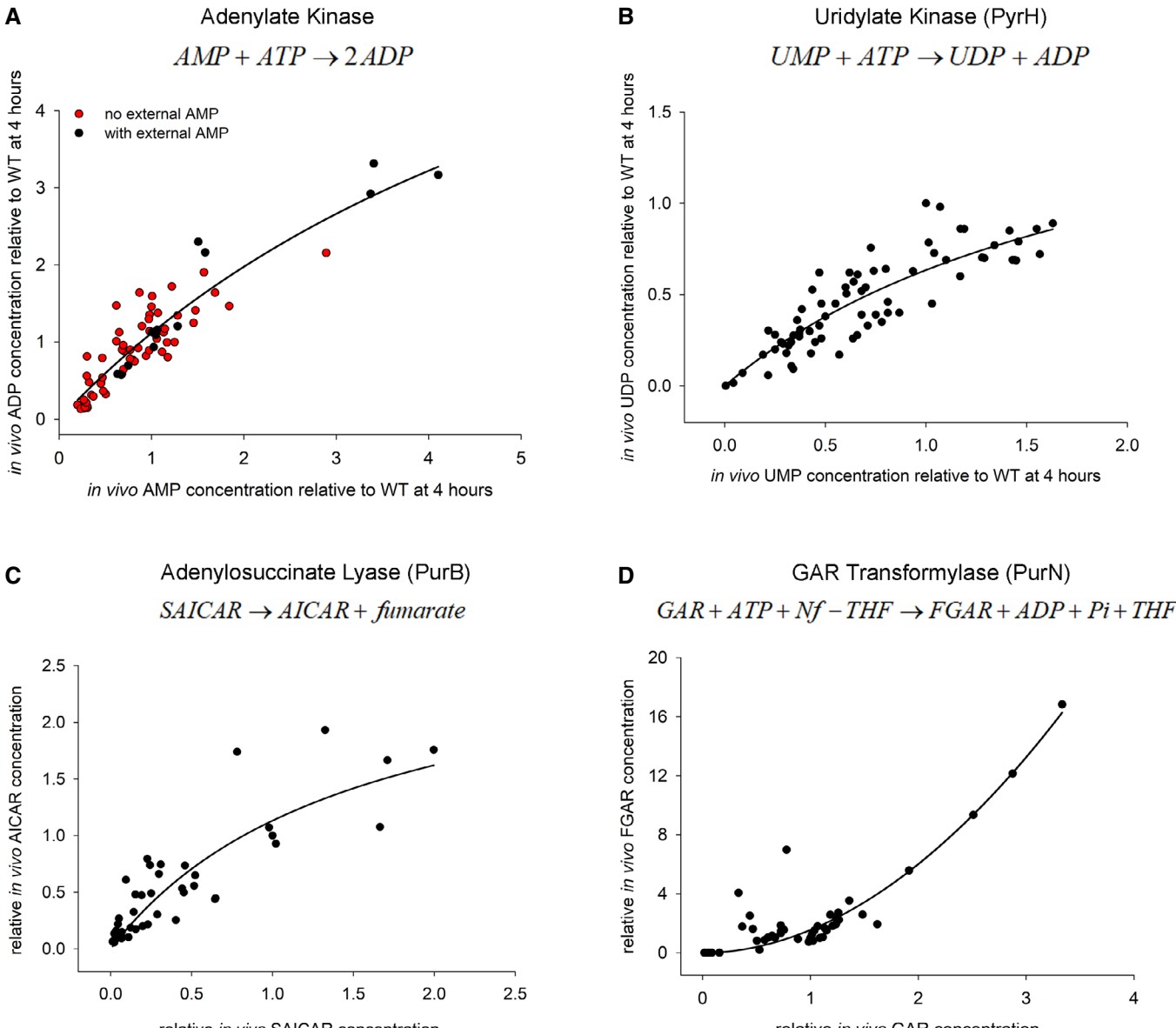

**Figure 7. In vivo activity curves of representative E. coli enzymes show hyperbolic or cooperative behavior.**

A–D   Steady-state levels of different metabolites obtained from metabolomics were used to derive activity curves for (A) adenylate kinase (AK), (B) uridylate kinase (PyrH), (C) adenylosuccinate lyase (PurB), and (D) GAR transformylase (PurN). AK, PyrH, and PurB appear to follow hyperbolic MM kinetics *in vivo*, while PurN shows Hill-like dependence with cooperativity. In panel (A), the red data points were acquired during growth in amino acid supplemented M9 medium in the absence of external AMP and are derived from WT and various stabilizing and destabilizing point mutations of AK (Adkar *et al*, 2019) as well as DHFR mutants used in the present study, while black points were derived exclusively from cultures of AK mutants supplemented with 1 mM external AMP (Adkar *et al*, 2019). Data points for all other panels (B-D) are derived from mutant DHFR strains in the present study. Since SAICAR, AICAR, GAR, and FGAR levels were undetectable in WT, all data points in (C-D) are relative to those observed in I91L+W133V mutant following 4 h of growth at 40°C. Data points in all panels represent metabolite levels for individual biological replicates without averaging.

Source data are available online for this figure.

drop in folate end products due to mutations in DHFR directly affects dTMP levels. However, we reasoned that since folate products are utilized in both *de novo* purine and pyrimidine biosynthesis pathways, several intermediates in those pathways might show varying levels of accumulation. Due to inherent low stability of certain intermediates, we could only detect a handful of metabolites

with confidence. Despite that, using our metabolomics data, we could successfully construct activity curves for three more enzymes: uridylate kinase (PyrH) from the *de novo* pyrimidine biosynthesis pathway, and adenylosuccinate lyase (PurB) and GAR transformylase (PurN) from the *de novo* purine biosynthesis pathway. As shown in Fig 7B and C, PyrH and PurB show a hyperbolic,

Michaelis–Menten-like dependence of product on substrate levels, like adenylate kinase. Strikingly, PurN (Fig 7D) shows a long lag followed by steep transition and fitted Hill-like kinetics with high significance ($P < 0.0001$ for MM as null model), much like Tmk, with an estimated Hill coefficient of ~2. However, since the substrate GAR (glycineamide ribonucleotide) for this enzyme is not commercially available, we were unable to test the kinetics of PurN in the presence of GAR supplementation.

We therefore conclude that while enzymes like Tmk and PurN show Hill-like kinetics *in vivo*, there are many that retain traditional MM kinetics in the cell. Presumably, this effect is dictated by the complex cellular milieu, on which we elaborate more in the Discussion.

## Discussion

Metabolic networks of cells are inherently intertwined, with substrates and products of one pathway being utilized by another pathway. As a result, perturbations produced in one pathway can easily percolate into others, usually magnifying the effects. The folate pathway or the 1-carbon metabolism pathway is a classic example of this, as reduced folates act as 1-carbon donors during biosynthesis of purines, pyrimidines, and amino acids. Kwon et al (2008) showed that for inhibition of DHFR activity using trimethoprim, accumulation of substrate dihydrofolate (DHF), in turn, results in inhibition of another downstream enzyme critical to folate metabolism: folylpoly-gamma-glutamate synthetase (FP-gamma-GS), in a domino-like effect (falling DHFR activity triggers a fall in the other enzyme's activity too). In this work, we show that in *E. coli* strains that harbor destabilizing mutations in *folA* gene, reduced DHFR activity strongly affects, among other factors, the pyrimidine biosynthesis pathway by reducing production of dTMP from dUMP via thymidylate synthase (ThyA) that uses a derivative of THF as one-carbon source. Much like a domino effect, such drop in dTMP levels due to mutations in DHFR results in a precipitous drop in dTDP/dTTP, mainly due to the strong cooperative *in vivo* activity of another downstream essential enzyme thymidylate kinase (Tmk) in the pyrimidine biosynthesis pathway. Drop in dTTP level eventually leads to an imbalance in the levels of deoxynucleotides, presumably causing errors in DNA replication, SOS response, and filamentation due to blocked cell division (Fig 6E).

An important finding from the current study is that enzymes can exhibit different kinetics *in vivo* depending on the source of the substrate. In case of Tmk, the enzyme showed a conventional Michaelis–Menten type *in vivo* activity when dTMP was externally supplied through the growth medium. However, when an equivalent concentration of dTMP was produced by the cell itself using its own cascade of enzymes in the pathway, it showed a dramatically different cooperative (Hill-like) activity. There are two important questions that arise out of this observation: First, why is the intrinsic *in vivo* activity of Tmk Hill-like? Second, what causes this shift from Hill-like to Michaelis–Menten (MM)? One of the most straightforward reasons for Hill-like enzyme activity is allosteric substrate binding. However, purified Tmk *in vitro* shows perfect MM kinetics, ruling out any intrinsic allostery of the enzyme. We also found that even in the presence of high concentrations of dUMP and dCTP (the

known inhibitors of Tmk), the activity of Tmk conforms to MM kinetics ruling out these metabolites as allosteric regulators (Fig EV3D). There are several other metabolites (GTP, GDP, GMP, dATP, UTP) in our targeted metabolomics analyses that are highly upregulated in the mutant strains (Fig 2C); however, they were upregulated in the presence of external dTMP as well (condition with MM-like kinetics) (Fig 5B), thereby ruling out their role in allosteric modulation of Tmk activity. The other possible mechanism of Hill-like kinetics is "limited diffusion" of one or more of the interacting components of a reaction (Savageau, 1995, 1998). Conventional MM enzyme kinetics relies on the assumption of free diffusion, and hence, laws of mass action are obeyed. However, in case of limited diffusion, conditions of spatial uniformity are no longer maintained; hence, law of mass action is not applicable. Theoretical work and simulations (Liebovitch *et al*, 1987; Kopelman, 1988; Li *et al*, 1990; Savageau, 1995, 1998; Frank, 2013) have shown that such diffusion-limited reactions often exhibit kinetics with Hill-like coefficients that are significantly higher than 1 and often fractional (so-called fractal kinetics), similar to Hill coefficients obtained with our data (Fig 7C, Hill coefficient=2.5). It has also been postulated that biological reactions, especially those that happen in dimensionally restricted environments like 1D channels or 2D membranes, exhibit fractal-like kinetics (Liebovitch *et al*, 1987; Schnell & Turner, 2004). In our case, it seems more reasonable that it is the substrate dTMP that has limited diffusion rather than the enzyme Tmk itself, since addition of external dTMP alleviates the Hill-like effect (in Materials and Methods, we show a derivation of Hill-like enzyme kinetics assuming that only the substrate is diffusion-limited, using a power law formalism as developed by Savageau (Savageau, 1995)). But why should dTMP be diffusion-limited? Substantial work in the recent past has shown that metabolic enzymes of a pathway, including those involved in purine biosynthesis (An *et al*, 2008; French *et al*, 2016) as well as in 1-carbon metabolism (Bhattacharyya *et al*, 2016), form a metabolon, a supramolecular complex comprised of transiently interacting enzymes, that allows efficient channeling of metabolites. Though channels help in easy exchange of metabolites between consecutive enzymes and prevent their unwanted degradation or toxicity in the cytosol, they have reduced dimensionality compared to the cytosol, thereby making motion less "random" and hence limiting diffusion of the substrate/products. In our case, it is possible that enzymes of the pyrimidine biosynthesis pathway as well as the salvage pathway form a metabolon, that limits diffusion of dTMP. External dTMP, on the other hand, is free to diffuse in the cytoplasm and hence results in traditional MM kinetics to emerge with a Hill coefficient of 1.

Though we do not have direct evidence of Tmk being involved in a metabolon, our study does show some circumstantial evidence. Our previous work on DHFR showed that toxicity and filamentation upon overexpression might be a hallmark of metabolon proteins (Bhattacharyya *et al*, 2016), through sequestration of neighboring/sequential proteins in the pathway. On a similar note, we found that overexpression of Tmk in WT *E. coli* cells is toxic and leads to filamentation (Fig EV5A and B), strongly suggesting that Tmk might be part of a metabolon. It is worth mentioning at this point that Tmk overexpression in mutant DHFR cells does not rescue filamentation (Fig EV5A). This is consistent with the metabolon hypothesis, since dTMP produced by the cell would still be confined to the metabolon,

and hence, overexpressed Tmk cannot overcome the problem of diffusion limitation of dTMP.

PurN is another enzyme that shows strong Hill-like kinetics in our study, and it is worth noting that the mammalian analog of PurN (the GAR Transformylase) constitutes one of the core proteins of the purinosome (Deng *et al*, 2012). Though in the absence of direct metabolic complementation studies, it is difficult to pinpoint which metabolite might be diffusion-limited, our study nevertheless strongly points toward the existence of a similar purinosome complex in *E. coli* as well. This is also indirectly supported by the fact that overexpression of PurN is highly toxic in *E. coli* (Kitagawa *et al*, 2005). Regarding adenylate kinase (AK), though several enzymes use up ATP, AMP, and ADP, AK as such is not part of a pathway, which might be the reason behind lack of any substantial cooperativity *in vivo*. Previous data from co-IP experiments also showed no interaction with other *E. coli* proteins (Bhattacharyya *et al*, 2016). Moreover, data from the ASKA library as well as our own (Fig EV5; Kitagawa *et al*, 2005; Bhattacharyya *et al*, 2016) show that overexpression of AK is not toxic, again supporting non-involvement of AK in any kind of metabolon. Of course, not every enzyme in a pathway is part of a metabolon, as presumably exemplified by data from PyrH and PurB which are, respectively, parts of the *de novo* pyrimidine and purine biosynthesis pathways yet show hyperbolic *in vivo* kinetics. Though PurB is part of the mammalian purinosome, it is not part of the core complex and is known to associate with weak interaction (Deng *et al*, 2012). Data from ASKA library show that PyrH is neutral upon overexpression (Kitagawa *et al*, 2005), though PurB is mildly toxic. PurB however catalyzes a second reaction outside the purine biosynthesis pathway, which might contribute to the observed toxicity.

For decades, enzyme activity has been studied *in vitro* with purified enzyme in dilute solution with excess substrate. Though *in vitro* measured parameters have been largely successful to interpret cellular data (Rodrigues *et al*, 2016; Adkar *et al*, 2017), in other cases they have only provided limited information (van Eunen *et al*, 2012). Substantial efforts in the recent past have therefore been directed toward replicating *in vivo* like conditions with purified enzymes (Garcia-Contreras *et al*, 2012; Davidi *et al*, 2016; Zotter *et al*, 2017). These include macromolecular crowding, pH conditions, and buffer capacity (van Eunen & Bakker, 2014). In this work, we present a potentially powerful general method for detecting *in vivo* activity of an enzyme using product to substrate ratios obtained from metabolomics and show how the *in vivo* activity curves of an enzyme can be markedly different (in case of Tmk strongly Hill-like) from its perfect MM-like kinetics *in vitro*. Based on available literature and some of our preliminary experiments/observations on Tmk and other proteins, metabolon formation and subsequent diffusion limitation of one of the substrates (dTMP in case of Tmk) seems like the most probable mechanism. Future work will prove or disprove this hypothesis. However, regardless of the mechanism, this work provides convincing evidence that the cellular environment can modulate enzyme activity in a very fundamental way, which explains a key bacterial phenotype in our case.

In this study, we used metabolomics as the key tool to link molecular effects of mutations to phenotype and illustrate precise biochemical and biophysical mechanisms through which altered metabolite levels modulate bacterial phenotypes, in this case filamentation. Detailed metabolomics analysis allowed us to pinpoint the pathway and specific enzyme responsible for the phenotype and, surprisingly, it turned out to be far downstream from the mutant locus (folA). Furthermore, the culprit, Tmk, does not use products of the folate pathway as a cofactor. Nevertheless, it appears that perturbation of the folate pathway caused by mutations in DHFR propagated downstream in a domino-like manner to create a bottleneck in a specific metabolite dTDP triggering cellular SOS response and pronounced phenotypic effects manifested in altered cell morphology. Altogether, our results show how metabolomics can be used as a stepping stone from biophysical analysis of variation of molecular properties of enzymes to phenotypic manifestation of mutations and close the gap in the multi-scale genotype–phenotype relationship.

# Materials and Methods

### Strains and media

The mutant DHFR strains chosen for this study (W133V, I91L+W133V, V75H+I155A, and V75H+I91L+I155A) were a subset of strains generated in and described in Bershtein *et al* (2012). Briefly, using structural and sequence analyses, positions were chosen that were buried in the protein and located at least 4 Å away from the active site, so that mutations introduced at these positions would have minimal effect on enzymatic activity. The identity of the mutations was selected such that they are appear with very low frequency in a multiple sequence alignment, and hence were intended to destabilize the protein, as was confirmed by stability measurements of the purified proteins (see table below). The single mutants were in most cases mild to moderately destabilizing, and hence, certain mutations were combined to increase the range of destabilization achieved. The following table summarizes the results of stability measurements on the single mutants (from Bershtein *et al*, 2012). However, the multiple site mutants could not be purified to quantities that were sufficient for *in vitro* biophysical and biochemical characterization.

| Mutant | Interaction | $T_m$ (°C) | $\Delta G$ (kcal/mol) | $\Delta\Delta G$ (kcal/mol) |
|--------|-------------|-----------|-----------|------------|
| WT | | 51.7 | −4.4 | 0 |
| V75H | Steric clash | 40.6 | −2.18 | 2.22 |
| I91L | Steric clash | 41.4 | −2.69 | 1.71 |
| W133V | Cavity formation due to large to small substitution | ND | −1.53 | 2.87 |
| I155A | Cavity formation due to large to small substitution | 38.5 | −2.42 | 1.98 |

Single as well as multiple site mutations were eventually introduced into the chromosomal copy of the *folA* at its endogenous locus keeping its regulatory region intact, and the effect of the mutations on its growth and morphology was measured at 30, 37, and 42°C. While WT, W133V, and V75H+I155A could grow up to 42°C,

mutants I91L+W133V and V75H+I91L+I155A could only survive up to 40°C, and hence, these strains were grown at 40°C instead of 42°C. The reason for choosing a wide range of temperature instead of following a single conventional temperature of 37°C was that *E. coli* is a gut bacterium and inhabits hosts whose core body temperatures span a large range (37–38°C for mammals, 40–45°C for birds (Guenther *et al*, 2010; Torre-Bueno, 1976)). Moreover, since the chosen mutants were temperature-sensitive, the phenotypic manifestation of the mutations was the largest at the extremes of temperatures, in case of *E. coli* at 42°C.

Wherever mentioned, M9 minimal medium *without amino acids* was only supplemented with 0.2% glucose and 1 mM MgSO$_4$, while M9 media *with amino acids* was supplemented with 0.2% glucose, 1 mM MgSO$_4$, 0.1% casamino acids, and 0.5 μg/ml thiamine. Casamino acids is a commercially available mixture of all amino acids except tryptophan, and cysteine is present in a very small amount. Wherever mentioned, 1 mM GMP was used as a source of purine, while dTMP (thymidine monophosphate) was used at a concentration of 0.25–1 mM. Thymidine was used at two different concentrations of 0.5 and 1 mM.

## Growth conditions

All strains were grown overnight from a single colony at 30°C, and subsequently, the culture was diluted to a final OD$_{600}$ of 0.01 in the specified medium and allowed to grow for 16–18 h in Bioscreen C (Growth Curves, USA) at 30, 37, or 42°C. Growth curves were fit to a 4-parameter Gompertz equation as described in Bhattacharyya *et al* (2017) to derive growth parameters. Error bars were calculated as SEM of three biological replicates.

## Light microscopy

Cells were grown overnight at 30°C from a single colony in the specified medium, diluted 1/100, and grown at various temperatures for 4 h. For DIC images in Figs 1C and D, and EV1A, cells were pelleted, washed with PBS, and concentrated. DAPI staining (Molecular probes) was performed for 10 min at RT according to the manufacturer's instructions. 1 μl of a concentrated culture was then mounted on a slide and slightly pressed by a cover slip. DIC and DAPI images were obtained at room temperature by Nikon Ti Eclipse Microscope equipped with iXon EMCCD camera (Andor Technologies). For live phase contrast images and time-lapse experiments (Fig 4A), cells were mounted on supplemented M9 + 1.5% low melting agarose (Calbiochem) pads. Pads were then flipped on #1.5 glass dish (Willco Wells), and the images were acquired at room temperature with Zeiss Cell Observer microscope. For DIC images in Fig EV5A, cells were placed on agar pads and images were acquired with Zeiss Cell Discoverer microscope.

## Analysis of cell lengths

MicrobeTracker Suite [http://microbetracker.org/] (Sliusarenko *et al*, 2011) was used to obtain distributions of cell length for phase contrast images, and Zeiss Intellesis Module was used to analyze DIC images. On average, 500 cells were analyzed for each presented distribution. The cell lengths are represented in all figures as boxplots, where the boundaries of the box represent the 25$^{th}$ and 75$^{th}$ percentiles, the line inside the box represents the median of the distribution, and the whiskers represent the 10$^{th}$ and 90$^{th}$ percentiles, while the dots represent the 5$^{th}$ and 95$^{th}$ percentiles.

## Statistical analysis

In our experiments, cell lengths of *E. coli* were not normally distributed (Shapiro–Wilk test). Hence, non-parametric Mann–Whitney test was used to determine whether the median cell lengths of two samples were significantly different. In Fig 6C, fits to two different models, Michaelis–Menten and 3-parameter Hill, were compared using extra sum-of-squares F-test using GraphPad Prism software v9.0.0.

## Metabolomics

Cells were grown overnight at 30°C from a single colony in the specified medium, diluted 1/100, and re-grown. WT, WT+0.5 μg/ml Tmp, and W133V mutant were grown at 42°C, while mutant I91L+W133V was grown at 40°C. For time course experiment, aliquots were removed after 2, 4, 6, and 8 h, and metabolites were extracted as described in Bhattacharyya *et al* (2016). Briefly, the cells were washed 2 times with chilled 1× M9 salts, and metabolites were extracted using 300 μl of 80:20 ratio of methanol:water that had been pre-chilled on dry ice. The cell suspension was immediately frozen in liquid nitrogen followed by a brief thawing (for 30 s) in a water bath maintained at 25°C and centrifugation at 4°C at maximum speed for 10 min. The supernatant was collected and stored on dry ice. This process of extraction of metabolite was repeated two more times. The final 900 μl extract was spun down one more time, and the supernatant was stored in −80°C till used for mass spectrometry. Metabolite levels were averaged over 2–3 biological replicates as increasing sample size beyond 3 did not significantly change the SEM. In Figs 6 and 7, data points represent metabolite levels for all biological replicates without averaging.

## Expression of SOS response genes by qPCR

Cells were grown overnight at 30°C from a single colony in the specified medium, diluted 1/100, and grown at 37°C or 42°C for 4 h. Based on OD$_{600}$ of the cultures, a volume equivalent to $5 \times 10^8$ cells were spun down (assuming 1 OD$_{600}$ = $8 \times 10^8$ cells) and Protect Bacteria RNA Mini Kit (Qiagen) was used to extract total RNA as described in Bhattacharyya *et al* (2018). Following reverse transcription (Bhattacharyya *et al*, 2018), expression of *recA*, *recN*, and *sulA* genes was quantified using QuantiTect SYBR Green PCR kit (Qiagen) using the following primers:

| | |
|---|---|
| recA_fwd | ACAAACAGAAAGCGTTGGCG |
| recA_rev | AGCGCGATATCCAGTGAAAG |
| recN_fwd | TTGGCACAACTGACCATCAG |
| recN_rev | GACCACCGAGACAAAGAC |
| sulA_fwd | GTACACTTCAGGCTATGCAC |
| sulA_rev | GCAACAGTAGAAGTTGCGTC |

As it was difficult to find a reference gene that would be expressed to similar levels in WT versus mutant DHFR strains, we used total RNA to normalize the expression levels.

Expression levels reported are average of 3 biological replicates. Error bars in Fig 4 represent 12% of the mean value.

## Tmk protein purification

The *tmk* gene was cloned in pET28a plasmid between *NdeI* and *XhoI* sites with an N-terminal histag. BL21(DE3) cells transformed with the plasmid were grown in Luria broth at 37°C till an OD of 0.6, induced using 1 mM IPTG and grown for an additional 5 h at 37°C. The protein was purified using Ni-NTA affinity columns (Qiagen) and subsequently purified by gel filtration using a HiLoad Superdex 75 pg column (GE). The protein was concentrated and stored in 10 mM potassium phosphate buffer (pH 7.2). The concentration of the proteins was measured by BCA assay (Thermo Scientific) with BSA as standard.

## Tmk activity assay

Tmk catalyzes the following reaction $dTMP + ATP \rightleftharpoons dTDP + ADP$, and the activity assay was carried out using the spectrophotometric assay as described in Nelson and Carter (1969). Briefly, the reaction mixture contained 5 mM $MgCl_2$, 65 mM KCl, 350 μM phosphoenolpyruvate (PEP), and 300 μM NADH. To obtain $K_M$ for dTMP, ATP concentration was fixed at 1 mM, while dTMP concentration was varied from 10 to 500 μM. The reaction mix without enzymes was incubated at 25°C for 5 min, and the reaction was initiated by adding 100 nM Tmk (final concentration) and 2 units of pyruvate/lactate dehydrogenase. The kinetic traces were recorded for every 5 s for a total time of 1 min. The data corresponding to the first 20 s were fitted to a linear model to obtain initial rates. To obtain $K_I$ of dCTP for Tmk, ATP and dTMP concentrations were fixed at 100 μM and 1 mM, respectively, while dCTP concentration was varied from 0.5 to 7.5 mM. Since conversion of dUMP to dTDP also produces ADP, the $K_I$ of dUMP could not be estimated by the spectrophotometric method. Instead, dTDP amounts produced in the reaction were determined by LC-MS. For the reaction, ATP and dTMP concentrations were fixed at 1 mM and 100 μM, respectively, while dUMP concentration was varied from 0.25 to 5 mM. The reaction was quenched at 40 s using 80% MeOH. The resulting samples were subjected to LC-MS analysis to obtain dTDP levels. For Fig EV3B and D, LC followed by mass spectrometry was used to directly measure dTDP levels.

## Derivation of the relationship between steady-state metabolite concentrations and kinetics of the enzyme *in vivo*

### Case I: Sequential enzymes in pathway with Michaelis–Menten kinetics

The following is an example of a pathway where several enzymes work sequentially:

$$E_1 + S_1 \underset{k_{-1}}{\overset{k_1}{\rightleftharpoons}} E_1 S_1 \underset{k_{-2}}{\overset{k_2}{\rightleftharpoons}} E_1 + S_2$$

$$E_2 + S_2 \underset{k_{-3}}{\overset{k_3}{\rightleftharpoons}} E_2 S_2 \underset{k_{-4}}{\overset{k_4}{\rightleftharpoons}} E_2 + S_3 \rightleftharpoons \ldots$$

For the pathway at steady state, the concentrations of the reactants, products, and intermediates do not change with time.

Therefore,

$$\frac{d[E_1 S_1]}{dt} = k_1[E_1][S_1] + k_{-2}[E_1][S_2] - (k_2 + k_{-1})[E_1 S_1] = 0 \quad (4)$$

$$\frac{d[E_2 S_2]}{dt} = k_3[E_2][S_2] + k_{-4}[E_2][S_3] - (k_4 + k_{-3})[E_2 S_2] = 0 \quad (5)$$

The enzyme concentrations can be written as:

$$[E_1] = [E_1]_0 - [E_1 S_1]$$
$$[E_2] = [E_2]_0 - [E_2 S_2] \quad (6)$$

$[S_1]$, $[S_2]$ are the steady-state concentrations of two sequential substrates (or products) in the pathway. Based on equations (4–6), we deduce:

$$[E_1 S_1] = \frac{(k_1[S_1] + k_{-2}[S_2])[E_1]_0}{k_{-1} + k_2 + k_1[S_1] + k_{-2}[S_2]} \quad (7)$$

$$[E_2 S_2] = \frac{(k_3[S_2] + k_{-4}[S_3])[E_2]_0}{k_{-3} + k_4 + k_3[S_2] + k_{-4}[S_3]} \quad (8)$$

Again, at steady state,

$$\frac{d[E_1]}{dt} = 0 = -k_1[E_1][S_1] + k_{-1}[E_1 S_1] - k_{-2}[E_1][S_2] + k_2[E_1 S_1] \quad (9)$$

Hence,

$$[E_1] = \frac{(k_2 + k_{-1})[E_1 S_1]}{k_1[S_1] + k_{-2}[S_2]} \quad (10)$$

Similarly, one can show that:

$$[E_2] = \frac{(k_4 + k_{-3})[E_2 S_2]}{k_3[S_2] + k_{-4}[S_3]} \quad (11)$$

Since the pathway is at steady state, concentrations of reactants and products of every reaction remain unchanged with time, hence.

$$\frac{d[S_2]}{dt} = k_2[E_1 S_1] - k_{-2}[E_1][S_2] + k_{-3}[E_2 S_2] - k_3[E_2][S_2] = 0 \quad (12)$$

Using the expressions of $[E_1]$, $[E_2]$ $[E_1 S_1]$ and $[E_2 S_2]$ from equations (7, 8, 10 and 11) into equation 12,

$$[E_1 S_1]\left[k_2 - \frac{k_{-2}[S_2](k_2 + k_{-1})}{k_1[S_1] + k_{-2}[S_2]}\right] + [E_2 S_2]\left[k_{-3} - \frac{k_3[S_2](k_4 + k_{-3})}{k_3[S_2] + k_{-4}[S_3]}\right] = 0$$

$$[E_1]_0 \frac{k_1 k_2[S_1] - k_{-1}k_{-2}[S_2]}{k_{-1} + k_2 + k_1[S_1] + k_{-2}[S_2]} = [E_2]_0 \frac{k_3 k_4[S_2] - k_{-3}k_{-4}[S_3]}{k_{-3} + k_4 + k_3[S_2] + k_{-4}[S_3]} \quad (13)$$

Using $K_{M1} = \frac{k_2 + k_{-1}}{k_1}$, $K_{M2'} = \frac{k_2 + k_{-1}}{k_{-2}}$, $K_{M2} = \frac{k_4 + k_{-3}}{k_3}$, $K_{M3} = \frac{k_4 + k_{-3}}{k_{-4}}$, where $K_{M1}$ is the Michaelis constant of $E_1$ for $S_1$, $K_{M2'}$, is that of $E_1$ for $S_2$, $K_{M2}$ is that of $E_2$ for $S_2$, $K_{M3}$ is that of $E_2$ for $S_3$, equation (13) can be written as:

$$[E_1]_0 \frac{k_2[S_1]/K_{M1} - k_{-1}[S_2]/K_{M2'}}{1 + [S_1]/K_{M1} + [S_2]/K_{M2'}} = [E_2]_0 \frac{k_4[S_2]/K_{M2} - k_{-3}[S_3]/K_{M3}}{1 + [S_2]/K_{M2} + [S_3]/K_{M3}}$$

(14)

Assuming that $K_{M2'}, K_{M3} \gg K_{M1}, K_{M2}$ (in other words if $k_{-2}$ and $k_{-4}$ are very small) or the products have very low affinity back toward the enzyme, equation (14) reduces to the following:

$$[E_1]_0 \frac{k_2[S_1]}{K_{M1} + [S_1]} = [E_2]_0 \frac{k_4[S_2]}{K_{M2} + [S_2]}$$

(15)

Equation (15) can be rearranged to get the following hyperbolic or Michaelis–Menten-like dependence of $S_2$ on $S_1$:

$$[S_2] = \frac{k_2 K_{M2}[E_1]_0[S_1]}{k_4 K_{M1}[E_2]_0 + (k_4[E_2]_0 - k_2[E_1]_0)[S_1]} = \frac{A[S_1]}{B + C[S_1]},$$

(16)

### Case II: Sequential enzymes in pathway with Hill-like kinetics

For a single enzyme, initial rate.

$$\nu_0 = \frac{\nu_{\max}[S]^m}{K_M + [S]^m},$$

(17)

where "$m$" is the Hill coefficient.

Now again consider the following scheme of sequential enzymes:

$$E_1 + S_1 \underset{k_{-1}}{\overset{k_1}{\rightleftharpoons}} E_1 S_1 \underset{k_{-2}}{\overset{k_2}{\rightleftharpoons}} E_1 + S_2$$

$$E_2 + S_2 \underset{k_{-3}}{\overset{k_3}{\rightleftharpoons}} E_2 S_2 \underset{k_{-4}}{\overset{k_4}{\rightleftharpoons}} E_2 + S_3 \rightleftharpoons \dots$$

For reactants and products to be at steady state, earlier we derive equation (15), which essentially equates the rate of production and consumption of $S_2$ through the two enzymes (with the assumption that $k_{-2}$ and $k_{-4}$ are very small). In such a situation if either $E_1$ or both $E_1$ and $E_2$ have Hill-like kinetics, equation (15) can be written as the following based on equation (17):

$$[E_1]_0 \frac{k_2[S_1]^m}{K_{M1}^* + [S_1]^m} = [E_2]_0 \frac{k_4[S_2]^n}{K_{M2}^* + [S_2]^n}$$

(18)

where $m$ and $n$ are the Hill coefficient analogs of the two consecutive enzymatic steps.

Equation (18) can be rearranged as:

$$[S_2]^n = \frac{k_2 K_{M2}^*[E_1]_0[S_1]^m}{k_4 K_{M1}^*[E_2]_0 + (k_4[E_2]_0 - k_2[E_1]_0)[S_1]^m} = \frac{A[S_1]^m}{B + C[S_1]^m}$$

(19)

Therefore,

$$[S_2] = \left[\frac{A[S_1]^m}{B + C[S_1]^m}\right]^{1/n}$$

(20)

A numerical solution of equation (20) hows that $S_2$ shows positive cooperativity as a function of $S_1$ only if $m > n$ (see Fig EV6).

### Power law formalism for fractal kinetics

For a system where reactants and products diffuse freely, the rate constant of the reaction is time-independent. However, under conditions of diffusion limitation (fractal kinetics), rate constant is no longer a constant, but varies with time in the following way:

$$k = k_0 t^{-h}$$

where $h$ is related to the fractal dimension of the medium.

In the following section, we adopt the power law formalism as shown in Savageau (1995) to convert the time-dependent rate constant to a time-independent one.

Considering the following simple reaction of two molecules of A forming a homodimer under conditions of diffusion limitation:

$$A + A \overset{k}{\longrightarrow} product$$

$$rate = \frac{d[A]}{dt} = -k(t)[A]^2 = -k_0 t^{-h}[A]^2$$

(21)

Integrating the above equation, we get [A] as a function of time.

$$[A] = \frac{1 - h}{k_0 t^{1-h}}$$

(22)

Rearranging this, we get.

$$t = \left(\frac{1 - h}{[A]k_0}\right)^{\frac{1}{1-h}}$$

(23)

In the next step, we replace $t$ in equation (21) with (23) to get:

$$Rate = -(1 - h)^{\frac{h}{h-1}} k_0^{\frac{1}{1-h}} [A]^{2 + \frac{h}{1-h}} = -k^*[A]^\alpha$$

(24)

where $\alpha = 2 + \frac{h}{1-h}$ (note that though this is a bimolecular reaction, the actual molecularity is >2 under fractal conditions).

### Application for enzyme kinetics

Assume the following simple case:

$$E + S \underset{k_{-1}}{\overset{k_1}{\rightleftharpoons}} ES \overset{k_2}{\longrightarrow} E + P.$$

Consider that the substrate S is diffusion-limited, hence $k_1$ (a bimolecular rate constant) will be time-dependent.

$$\frac{d[ES]}{dt} = k_1(t)[E][S] - (k_{-1} + k_2)[ES] = k_1^*[E][S]^n - (k_{-1} + k_2)[ES].$$

where $k_1^*$ is the apparent time-independent rate constant and $n$ is related to the fractal dimension of the medium.

At steady state, $\frac{d[ES]}{dt} = 0$.

Hence, $[ES] = \frac{[E]_0[S]^n}{\left(\frac{k_2 + k_{-1}}{k_1^*}\right) + [S]^n}$

Rate of the reaction, $\nu = k_2[ES] = \dfrac{k_2[E]_0[S]^n}{\left(\frac{k_2+k_{-1}}{k_1^*}\right)+[S]^n} = \dfrac{k_2[E]_0[S]^n}{K_M^*+[S]^n}$

where $K_M^*$ is the apparent Michaelis constant and $n$ is the Hill coefficient analog.

## Data availability

The datasets produced in this study are available in the following database:

Metabolomics data: Metabolights MTBLS2795 (www.ebi.ac.uk/metabolights/MTBLS2795). The processed data are included in source data for Figs 2 and 5.

**Expanded View** for this article is available online.

## Acknowledgement
This work was supported by NIH grants 1RO1 GM068670 and 1R35 GM139571 to E.I.S.

## Author contributions
Conceptualization, SBh, SBe, and EIS; Methodology, SBh, SBe, BVA, JW, and EIS; Formal Analysis, SBh, SBe, BVA, and EIS; Investigation, SBh, SBe, BVA, JW; Writing-original draft, SBh, and EIS; writing-review & editing, SBh, SBe, BVA, and EIS; Supervision, EIS; Funding acquisition, EIS.

## Conflict of interest
The authors declare that they have no conflict of interest.

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
