## [Review Process File · Molecular Systems Biology]

Metabolic response to point mutations reveals principles of modulation of in vivo enzyme activity and phenotype

Sanchari Bhattacharyya, Shimon Bershtein, Bharat Adkar, Jaie Woodard, and Eugene Shakhnovich
DOI: [10.15252/msb.202110200](https://doi.org/10.15252/msb.202110200)

Corresponding author(s): Eugene Shakhnovich (eugene@belok.harvard.edu) , Sanchari Bhattacharyya (sbhattacharyya@fas.harvard.edu)

Review Timeline:

Submission Date:	1st Jan 21
Editorial Decision:	4th Feb 21
Revision Received:	13th Apr 21
Editorial Decision:	28th Apr 21
Revision Received:	8th May 21
Accepted:	11th May 21

Editor: Jingyi Hou

Transaction Report:

Thank you for submitting your work to Molecular Systems Biology. We have now heard back from the three reviewers who agreed to evaluate your study. As you will see below, the reviewers acknowledge that the presented findings and data are potentially interesting. They raise, however, a series of concerns, which we would ask you to address in a major revision.

The reviewers' recommendations are rather straightforward and there is no need to reiterate their comments. In light of the concerns of Reviewers #2 and #3, we would ask you to edit the manuscript to make sure that the main findings are sufficiently clear and easily accessible to the general audience of Molecular Systems Biology. Reviewer #3 pointed out that the mechanisms underlying the different Tmk kinetics in vitro versus in vivo remain unclear. This reviewer provides constructive suggestions to strengthen the findings in this regard.

All other issues need to be satisfactorily addressed. As you may already know, our editorial policy allows in principle a single round of major revision, so it is essential to provide responses to the reviewers' comments that are as complete as possible. Please feel free to contact me in case you would like to discuss in further detail any of the issues raised by the reviewers.

On a more editorial level, please do the following.

REFEREE REPORTS

Reviewer #1:

In this manuscript, Bhattacharyya et al. demonstrate the mechanistic links between a phenotypic event, filamentation, with a genotypic event, a genetic sequence variant. This work does so through a series of metabolomic experiments and kinetic modeling analysis with *E. coli* K-12 MG1655. The manuscript provides a detailed analysis working to link the genotype and phenotype, a current significant challenge in biology, by leveraging the depth and breadth of information provided through kinetic modeling and metabolomics. The interpretation of the results considers several

metabolic and biochemical mechanisms that may contribute together towards the phenotype: 1) the reduction DHFR activity through destabilizing mutations, 2) the increase in DHF substrate leading to downstream inhibition of enzyme FP-gamma-GS, 3) the cooperative in vivo activity between enzymes DHFR and TmK, and 4) the role of metabolite channeling via metabolons relative and their potential effect on the general availability of metabolites in the cytosol. The results additionally revealed the possibility of TmK exhibiting different enzyme activity, whether in vitro or in vivo, which has several implications for practice. Many studies hypothesize the phenotypic effect of sequence variants and test these only through growth assays, neglecting the link's mechanistic aspect. Those researching the genotype to phenotype link, such as systems biologists, should find this work of great interest.

The below are some minor points that I believe would help flesh out some gaps in the manuscript's contents:

The Strain and media Methods section describes the growth and morphology of mutants being measured for 30, 37 and 42C, though Figure 1 and 6 also includes these measurements with 40C. The presence of measurements at 40C is less frequent than the other temperatures. Why is the condition of 40C included, though less frequently than the other temperatures? I'm guessing that mutants W133V and V75H+I155A could survive up to 42C while V75H+I155A and V75H+I91L+I155A could only survive to 40C. If this is already explained within one of the methods section and I have missed it, please place it earlier in the methods section.

This manuscript works to explain mechanistic properties at multiple scales to understand the genotype to phenotype relationship. However, the mechanisms describing the stability changes due to the substitutions aren't explained in the same detail. The authors are likely relying on the referenced studies of the same mutations to provide the additional details, though this stands out as a gap in the manuscript's content (Bershtein et al., 2013; Bershtein et al., 2012)). Could the authors elaborate on the mechanisms of structural instability for single and combinations of mutations (such as residue side chains and steric effects) or give a summary of these details if they've already been covered in the previous studies for these mutations? The results describe unexpected variability between the combinations of substitutions, such as with Figure 4 where 3 substitutions (V75H+I91L+I155A) demonstrate lower SOS response signals than 2 substitutions (I91L+W133V) at 37C.

Some experiments often include the full set of mutants. According to the Methods section, it is expected that the increase in mutations should result in an increase in detrimental effects to the host. Counter to this expectation, those experiments that do include most/all mutants sometimes demonstrate a substantial difference in results between strains with 2 mutations. Why do the authors choose a particular mutant or subset of mutants in those experiments that don't consider all 4 strains?

The below are some trivial mistakes to correct:

Some of the symbols from equation (1) did not render properly with the PDF.

Line 179 "We found that a mutant E. coli has been knocked out for the e14 region does not filament upon TMP treatment (Figure 4D)" is unclear.

Line 797: "see Materials and Methods", the section is instead named "Methods" in this draft.

Line 252: "Cooperative of enzymatic activity of Tmk in vivo explains low dTDP levels" could be more

clear as "Cooperative enzymatic activity of Tmk in vivo explains low dTDP levels"

Reviewer #2:

In the manuscript "Metabolic response to point mutations reveals principles of modulation of in vivo enzyme activity and phenotype", the authors utilize several *E. coli* mutants, that have abnormal phenotypes under specific nutrition conditions, to study how metabolites influence enzyme activity. The authors conclude that the abnormal phenotype in mutant is due to an SOS response induced by lower dTDP/dTMP ratios, which influences pathway innate cooperativity of the thymidylate kinase (Tmk) enzyme. This cooperativity is concluded to occur in vivo, but not in purified enzyme. This is an important area of study and the authors are addressing an important question since how pathway flux is regulated lacks general principles. The logic of this study is kind of complicated and indirect, and requires some background knowledge of bacteria growth. I read over their cooperatively argument several times and had trouble understanding it. Are they saying that because the enzymes exist in a series there is cooperatively as a result of having multiple steps even if each enzyme is hyperbolic in its activity curve by itself? But their conclusion is cooperativity in Tmk which is a single step in the pathway. More explanation is needed. The author should also improve the presentation of results and conclusion for broader readers.

Comments

- Because of the indirect logic of experimental design (that is they construct the curves from independent measurements rather than directly moving along the curve), it is better to provide a diagram to show the outline: mutant of DHFR influence de novo synthesis of dTMP, and than influence dTDP/dTTP by some in vivo synergic effect of Tmk, which than triggers SOS response to inhibit division, and forces bacteria to grow to a filament. The whole process is not intuitive for most readers.
- Figure 3A shows the imbalance of nucleotides is similar in M9 base media with that with amino acid. What is the reason for why the mutants in M9 base media without amino acids do not become filamentous? I guess it is because growth rate in base media is pretty low, the imbalance of nucleotides may not influence DNA synthesis, and thus does not trigger SOS responses. The author should clarify this in main text.
- Label in figure 4 is confusing: legend says lime color represents media with amino acid and dTMP in 37°C, but all subplots A-C include bar with this color labeled with 42°C.

Minor:

- Display of arrows in equation (1)

Reviewer #3:

Bhattacharya et al. study the metabolic origins of filamentation in DHFR-deficient *E. coli*. Filamentation is a documented side effect of trimethoprim treatment (Sagurdekar et al. (2011) BMC Genomics v12, 538), and while the biological/clinical relevance of the phenomenon is not entirely clear, it is often associated with thymine depletion, cell stress, and thymineless death (TLD). The authors seek to explain the molecular origins of this phenotype using a combination of growth rate measurements, filamentation measurements, metabolomics, and focused in vitro biochemical studies. More specifically, the authors show that filamentation:
1) depends on the addition of amino acids to the media, and is enhanced in DHFR LOF mutations by increasing temperature,

2) is rescued by the addition of WT DHFR, and is not a consequence of expression of the mutant proteins,

3) is associated with decreased dTMP and dTTP pools,

4) it is associated with upregulation of the recA/sulA SOS response,

5) it is often reversible and can be rescued by the addition of dTMP.

In addition to characterizing the conditions influencing filamentation, the authors show the thymidylate kinase (Tmk) seems to exhibit cooperative kinetics *in vivo*, and Michaelis-Menten like kinetics *in vitro*. As a consequence of the cooperative relationship between dTMP (substrate) and dTDP (product), intracellular pools of dTDP and dTTP show reduced recovery even as dTMP concentrations begin to increase, potentially exacerbating the conditions in which filamentation occurs. The observed discrepancy between the apparent kinetics *in vivo* and that measured *in vitro* suggests the possibility of Tmk competitive inhibition (or other regulation) *in vivo*, or, as the authors intriguingly put forth, Tmk participation in a larger metabolon or structured environment.

Major comments/concerns:

(1) Overall, the manuscript adds to our understanding of how folate metabolic deficiencies drive bacterial filamentation, though many aspects of the work seem to echo prior findings. For example, it has been known since the 70's that amino acid supplementation enhances cell death in response to trimethoprim by avoiding the bacteriostatic effects of amino acid starvation (Then and Ahgehrn. (1973) *Journal of Infectious Diseases* v. 128, S498, Amyes and Smith. (1974) *AAC* v.5 p.169). This result was expanded upon in more recent metabolomics work (Kwon et al (2010) *ACS Chem Biol* v.5 p.787). So the finding that filamentation (a marker of thymine limitation) depends on amino-acid supplementation is unsurprising. It is also established that DHFR LOF is associated with reductions in dTTP pools (Kwon et al 2010), and that thymine limitation in *E. coli* is associated with the SOS response (Sagurdekar et al. (2011) - this work also investigates the interactions with the deoCABD operon). Many of these previous works focused on growth or viability rather than filamentation as a phenotype, but the distinction seems a bit nuanced to me. Thus, I expect the studies on filamentation conditions and the role of recA will be of circumscribed interest to researchers studying bacterial folate metabolism and the response to trimethoprim.

(2) The finding that Tmk exhibits different kinetics *in vitro* vs. *in vivo* is IMO the most interesting part of the paper. It suggests caution in modeling intracellular processes using MM-like kinetics, and tantalizingly hints at TMK participation in a folate metabolon. It also establishes a potential procedure for detecting this phenomenon (using MS to investigate product/substrate ratios in the presence and absence of external substrate supplementation). However, the mechanism underlying the kinetics difference *in vivo* vs *in vitro* is unknown, and without an explanation for the observed data, one becomes modestly concerned that it could reflect some sort of experimental artifact. While I can certainly appreciate that identifying the mechanism requires significant further work, it would certainly add a huge amount of impact to the current story. The current manuscript (without a mechanism) could be strengthened in a few ways:

- The idea of the metabolon is very interesting, but is put forward with substantial force given the available data. While the data presented strongly indicate that competitive inhibition by dCTP/dUMP is not the origin of the effect, I wonder if it could instead be regulation or inhibition by some other metabolite accumulating in the mutant cell lines? Even allosteric (rather than competitive) inhibition? What would be/are there other compelling candidates?
- It would help to have additional experiments (positive and negative controls, of sorts) that establish expectations for what the data in fig 7 should look like for enzymes known to be (or not) in metabolons. If you take an enzyme not thought to be in a metabolon, can you show that it obeys MM kinetics whether substrate is supplied exogenously (or not)? If you take an enzyme known to

be in a metabolon, can you show that it behaves like Tmk - e.g. has the same Hill like dependence on substrate in the absence of external supplementation, and that this is lost in the presence of supplementation?

(3) The paper would benefit from an introduction that spends less time on the broad problem of mapping genotype to phenotype, but provides more context to the specific problem at hand. In particular:

- The authors cite heavily their own (nice) work on mapping biochemical parameters to phenotype, but should also mention other contributions to the rich literature on the topic, perhaps including work from Hartl (Dykhuizen et al (1987) Genetics v.115:25), and Anthony Dean (Lunzer et al (2005) Science v310:499)
- It would help to briefly review/provide context from the extensive prior literature on DHFR LOF, dTTP limitation, filamentation, and thymineless death
- The authors mention fractal kinetics but do not define/explain this term until much later

Minor concerns:

- dTMP vs TMP is at times easy to confuse. The authors might consider dTMP vs Tmp? Even changing capitalization might help readers to follow
- It is sometimes feels inconsistent that the authors compare results for only some mutants and some temperatures and don't always include a WT control for each condition
- The authors do not discuss the role of ATP in the reaction throughout the manuscript. While it seems clear from the mass spec data that ATP is not depleted relative to WT in the mutant cell lines, they should state this, and point out that their kinetics modeling assumes ATP is not limiting.
- In several places they refer to the cells having DNA damage (e.g. lines 62-63, p. 3). While I agree this is very likely, it is not *shown*, so the authors should be a bit careful with their language.
- On p.4/5 (section starting on Filamentous strains exhibit imbalance...) it would be helpful if the authors can explain their rationale for focusing/choosing two mutants out of those characterized in Fig. 1 and 2 for metabolomics
- On page 5, lines 122-123, the parenthetical listing of mutants and conditions characterized is confusing. Could the authors just enumerate these with a numbered list?
- In Fig. 5 the authors use one type of assay (time dependent colony counts, panels B,C) to show loss of viability/TLD at high doses of TMP, but a different assay (dilution spots + outgrowth) to show a lack of TLD for DHFR LOF mutations. It would help to show the dilutions spots and outgrowth for the TLD condition (WT at high TMP, +/- GMP) as a reference/control in panel 5D. Panels 5B/C also really need titles
- In Fig.2 - y axis needs units (presuming per minute as in Fig 1, but would be good to reiterate)
- P.9 line 248 - 20-fold overexpression, "overexpression" is not really the right word here, seems like over-abundance?
- P. 11 line 306, say TMK is Hill-like under native conditions. I think the native conditions are unknown (or at least not quantitatively defined) and depend on the growth environment of E. coli (mammalian gut?) It is probably more accurate to say presence or absence of exogenous dTMP.

First we would like to thank the reviewers for carefully reading our manuscript and extremely constructive and helpful comments.

Reviewer #1:

In this manuscript, Bhattacharyya et al. demonstrate the mechanistic links between a phenotypic event, filamentation, with a genotypic event, a genetic sequence variant. This work does so through a series of metabolomic experiments and kinetic modeling analysis with *E. coli* K-12 MG1655. The manuscript provides a detailed analysis working to link the genotype and phenotype, a current significant challenge in biology, by leveraging the depth and breadth of information provided through kinetic modeling and metabolomics. The interpretation of the results considers several metabolic and biochemical mechanisms that may contribute together towards the phenotype: 1) the reduction DHFR activity through destabilizing mutations, 2) the increase in DHF substrate leading to downstream inhibition of enzyme FP-gamma-GS, 3) the cooperative in vivo activity between enzymes DHFR and TmK, and 4) the role of metabolite channeling via metabolons relative and their potential effect on the general availability of metabolites in the cytosol. The results additionally revealed the possibility of TmK exhibiting different enzyme activity, whether in vitro or in vivo, which has several implications for practice. Many studies hypothesize the phenotypic effect of sequence variants and test these only through growth assays, neglecting the link's mechanistic aspect. Those researching the genotype to phenotype link, such as systems biologists, should find this work of great interest.

We thank the reviewer for the appreciative comments.

The below are some minor points that I believe would help flesh out some gaps in the manuscript's contents:

The Strain and media Methods section describes the growth and morphology of mutants being measured for 30, 37 and 42C, though Figure 1 and 6 also includes these measurements with 40C. The presence of measurements at 40C is less frequent than the other temperatures. Why is the condition of 40C included, though less frequently than the other temperatures? I'm guessing that mutants W133V and V75H+I155A could survive up to 42C while V75H+I155A and V75H+I91L+I155A could only survive to 40C. If this is already explained within one of the methods sections and I have missed it, please place it earlier in the methods section.

We thank the reviewer for pointing this out. Indeed, unlike WT and the other mutants, V75H+I155A and V75H+I91L+I155A mutant strains could only grow up to 40C. Hence, we chose this temperature. This information is now included in the "Strains and media" section of the Methods (Lines 788-790).

This manuscript works to explain mechanistic properties at multiple scales to understand the genotype to phenotype relationship. However, the mechanisms describing the stability changes due to the substitutions aren't explained in the same detail. The authors are likely relying on the

referenced studies of the same mutations to provide the additional details, though this stands out as a gap in the manuscript's content (Bershtein et al., 2013; Bershtein et al., 2012)). Could the authors elaborate on the mechanisms of structural instability for single and combinations of mutations (such as residue side chains and steric effects) or give a summary of these details if they've already been covered in the previous studies for these mutations? The results describe unexpected variability between the combinations of substitutions, such as with Figure 4 where 3 substitutions (V75H+I91L+I155A) demonstrate lower SOS response signals than 2 substitutions (I91L+W133V) at 37C.

We have now included the stability measurement data (from Bershtein *et al*, PNAS, 2012) of the concerned single site mutants used in this study in the Methods section along with a possible mechanism of structural instability (Lines 781-785). We also mention that the multi-site mutants could not be purified to sufficient amounts to carry out biophysical and biochemical characterization. Based on simple additivity of the folding free energies, one would naively expect that the triple mutant would be more destabilized compared to the double mutant and would hence elicit higher SOS response. However, it is possible that due to epistatic effects of mutations, the triple mutant is not as destabilized as expected, and hence lower SOS response.

Some experiments often include the full set of mutants. According to the Methods section, it is expected that the increase in mutations should result in an increase in detrimental effects to the host. Counter to this expectation, those experiments that do include most/all mutants sometimes demonstrate a substantial difference in results between strains with 2 mutations. Why do the authors choose a particular mutant or subset of mutants in those experiments that don't consider all 4 strains?

We agree that for some experiments (particularly metabolomics and cell viability assay), we only used WT, the single mutant W133V and the double mutant I91L+W133V. This was mainly because in most of our assays, W133V and the double mutant V75H+I155A behave similarly (in terms of growth rates, extent of filamentation and ability to grow up to 42°C), while I91L+W133V and the triple mutant V75H+I91L+I155A behaved similarly. Hence, we chose one representative example from each of these two groups. Moreover, despite being a double mutant, strain I91L+W133V had the lowest growth rate among all mutants, which makes it an ideal candidate to study the extreme effect of mutations.

The below are some trivial mistakes to correct:

Some of the symbols from equation (1) did not render properly with the PDF.

This has been corrected now.

Line 179 "We found that a mutant *E. coli* has been knocked out for the e14 region does not filament upon TMP treatment (Figure 4D)" is unclear.

We agree this statement was not clear and have revised it.

Line 797: "see Materials and Methods", the section is instead named "Methods" in this draft.

We have corrected this now.

Line 252: "Cooperative of enzymatic activity of Tmk in vivo explains low dTDP levels" could be more clear as "Cooperative enzymatic activity of Tmk in vivo explains low dTDP levels"

We have corrected this now.

Reviewer #2:

In the manuscript "Metabolic response to point mutations reveals principles of modulation of in vivo enzyme activity and phenotype", the authors utilize several *E. coli* mutants, that have abnormal phenotypes under specific nutrition conditions, to study how metabolites influence enzyme activity. The authors conclude that the abnormal phenotype in mutant is due to an SOS response induced by lower dTDP/dTMP ratios, which influences pathway innate cooperativity of the thymidylate kinase (Tmk) enzyme. This cooperativity is concluded to occur in vivo, but not in purified enzyme.

This is an important area of study and the authors are addressing an important question since how pathway flux is regulated lacks general principles. The logic of this study is kind of complicated and indirect and requires some background knowledge of bacteria growth. I read over their cooperatively argument several times and had trouble understanding it. Are they saying that because the enzymes exist in a series there is cooperativity as a result of having multiple steps even if each enzyme is hyperbolic in its activity curve by itself? But their conclusion is cooperativity in Tmk which is a single step in the pathway. More explanation is needed. The author should also improve the presentation of results and conclusion for broader readers.

We started out this section with the question whether steady state concentrations of metabolites obtained from metabolomics can truly provide any insight into the actual kinetics of the enzymes *in vivo*. Through the derivation (in the Appendix PDF), we show that for a series of enzymes in a pathway, if individual enzymes obey hyperbolic or MM kinetics, then the steady state concentrations of their substrate and product will also show the same hyperbolic behavior. On the other hand, if one or more enzyme in the pathway shows cooperative kinetics by itself, then the steady state levels of the concerned metabolites (substrate and product) will also show cooperativity, provided that the concerned enzyme is more cooperative than the next enzyme in the pathway ($m > n$). As a result, the Hill-like and hyperbolic curves obtained for Tmk from metabolomics data indeed allows us to infer about the kinetics of Tmk under these two conditions. We have

tried to address this concept in a clearer way now, so it is more accessible to the general audience (Lines 301-331).

Comments

- Because of the indirect logic of experimental design (that is they construct the curves from independent measurements rather than directly moving along the curve), it is better to provide a diagram to show the outline: mutant of DHFR influence de novo synthesis of dTMP, and then influence dTDP/dTTP by some in vivo synergic effect of Tmk, which then triggers SOS response to inhibit division, and forces bacteria to grow to a filament. The whole process is not intuitive for most readers.

We thank the reviewer for this important suggestion. Accordingly, we have now included a new panel Figure 6E that describes a flow chart highlighting the exact sequence of events that lead mutant DHFR strains to filament. In addition, we have aimed to summarize this in the early part of the Discussion too (Lines 407-409).

- Figure 3A shows the imbalance of nucleotides is similar in M9 base media with that with amino acid. What is the reason for why the mutants in M9 base media without amino acids do not become filamentous? I guess it is because growth rate in base media is pretty low, the imbalance of nucleotides may not influence DNA synthesis, and thus does not trigger SOS responses. The author should clarify this in main text.

This is an important point. It is true that even in M9 medium without added amino acids, the imbalance among deoxy nucleotides exists due to very low dTTP. However, as the reviewer has already mentioned, the cells grow very poorly under this condition primarily because Methionine and purine (IMP) levels are very low (Figure 2), hence protein and RNA synthesis are stalled (the so-called stringent response). In absence of duplication of the cytoplasm, the cell does not really proceed to duplicate its chromosome. Hence imbalance among deoxy nucleotides does not really manifest in terms of errors, hence there is no SOS response (Figure 3) and no filamentation. This is now clarified in the text (Lines 156-161).

- Label in figure 4 is confusing: legend says lime color represents media with amino acid and dTMP in 37°C, but all subplots A-C include bar with this color labeled with 42°C.

We have corrected this mistake in Fig 4 in the revised version.

Minor:

- Display of arrows in equation (1)

This has now been rectified.

Reviewer #3:

Bhattacharya et al. study the metabolic origins of filamentation in DHFR-deficient *E. coli*. Filamentation is a documented side effect of trimethoprim treatment (Sagurdekar et al. (2011) *BMC Genomics* v12, 538), and while the biological/clinical relevance of the phenomenon is not entirely clear, it is often associated with thymine depletion, cell stress, and thymineless death (TLD). The authors seek to explain the molecular origins of this phenotype using a combination of growth rate measurements, filamentation measurements, metabolomics, and focused in vitro biochemical studies. More specifically, the authors show that filamentation:

- 1) depends on the addition of amino acids to the media, and is enhanced in DHFR LOF mutations by increasing temperature,
- 2) is rescued by the addition of WT DHFR, and is not a consequence of expression of the mutant proteins,
- 3) is associated with decreased dTMP and dTTP pools,
- 4) it is associated with upregulation of the *recA/sulA* SOS response,
- 5) it is often reversible and can be rescued by the addition of dTMP.

In addition to characterizing the conditions influencing filamentation, the authors show the thymidylate kinase (Tmk) seems to exhibit cooperative kinetics in vivo, and Michaelis-Menten like kinetics in vitro. As a consequence of the cooperative relationship between dTMP (substrate) and dTDP (product), intracellular pools of dTDP and dTTP show reduced recovery even as dTMP concentrations begin to increase, potentially exacerbating the conditions in which filamentation occurs. The observed discrepancy between the apparent kinetics in vivo and that measured in vitro suggests the possibility of Tmk competitive inhibition (or other regulation) in vivo, or, as the authors intriguingly put forth, Tmk participation in a larger metabolon or structured environment.

Major comments/concerns:

(1) Overall, the manuscript adds to our understanding of how folate metabolic deficiencies drive bacterial filamentation, though many aspects of the work seem to echo prior findings. For example, it has been known since the 70's that amino acid supplementation enhances cell death in response to trimethoprim by avoiding the bacteriostatic effects of amino acid starvation (Then and Ahgehrn. (1973) *Journal of Infectious Diseases* v. 128, S498, Amyes and Smith. (1974) *AAC* v.5 p.169). This result was expanded upon in more recent metabolomics work (Kwon et al (2010) *ACS Chem Biol* v.5 p.787). So the finding that filamentation (a marker of thymine

limitation) depends on amino-acid supplementation is unsurprising. It is also established that DHFR LOF is associated with reductions in dTTP pools (Kwon et al 2010), and that thymine limitation in *E. coli* is associated with the SOS response (Sagurdekar et al. (2011) - this work also investigates the interactions with the deoCABD operon). Many of these previous works focused on growth or viability rather than filamentation as a phenotype, but the distinction seems a bit nuanced to me. Thus, I expect the studies on filamentation conditions and the role of recA will be of circumscribed interest to researchers studying bacterial folate metabolism and the response to trimethoprim.

We agree with the reviewer that loss in dTTP and filamentation are documented manifestations of TMP treatment and DHFR LOF mutation. However, first, we wish to argue that DHFR mutations in the present study are not LOF mutations in the true sense of the term, not just because they reduce net catalytic capacity by reducing protein abundance and not by reducing activity per se, but also because metabolically and phenotypically they mimic WT treated with sub-lethal concentrations of Trimethoprim. In other words, they seem to have a small amount of net residual DHFR activity that makes them quite different from a LOF mutation or treatment with high concentration of Trimethoprim. In fact, in our study, by measuring cell filamentation at different concentrations of Trimethoprim, we were able to show (to the best of our knowledge for the first time) a non-monotonic dependence of cell length on Trimethoprim concentration (Figure EV1 C,D), indicating that filamentation is very sensitive to the extent of DHFR inhibition achieved.

Second, previous studies (Sangurdekar et al (2011)) only reported filamentation under bactericidal conditions or TLD, which is Trimethoprim treatment under rich media conditions (LB and M9 supplemented with amino acids and purine). We show that cell filamentation can be completely reversible even in rich medium, provided there is only partial inhibition of DHFR activity. Cell death (TLD) happens only at *very high* concentrations of Trimethoprim in rich medium, and therefore this study disentangles the relation of DHFR activity levels with filamentation and TLD.

We have elaborated on these points in the text (Lines 190-199 and 210-227) and included a new panel in Figure 4 showing the extent of SOS response in media supplemented with amino acid and GMP.

(2) The finding that Tmk exhibits different kinetics in vitro vs. in vivo is IMO the most interesting part of the paper. It suggests caution in modeling intracellular processes using MM-like kinetics, and tantalizingly hints at TMK participation in a folate metabolon. It also establishes a potential procedure for detecting this phenomenon (using MS to investigate product/substrate ratios in the presence and absence of external substrate supplementation). However, the mechanism underlying the kinetics difference in vivo vs in vitro is unknown, and without an explanation for the observed data, one becomes modestly concerned that it could

reflect some sort of experimental artifact. While I can certainly appreciate that identifying the mechanism requires significant further work, it would certainly add a huge amount of impact to the current story. The current manuscript (without a mechanism) could be strengthened in a few ways:

- The idea of the metabolon is very interesting but is put forward with substantial force given the available data. While the data presented strongly indicate that competitive inhibition by dCTP/dUMP is not the origin of the effect, I wonder if it could instead be regulation or inhibition by some other metabolite accumulating in the mutant cell lines? Even allosteric (rather than competitive) inhibition? What would be/are there other compelling candidates?

We definitely cannot rule out at this point about cooperativity arising due to allosteric modulation by other metabolites accumulating in the cell. However based on our targeted metabolomics analyses, we find that aside from dUMP and dCTP which we already tested, several other metabolites (GTP, GDP, GMP, dATP, UTP) also accumulate in mutant DHFR strains (Figure 2C), however we find that they are upregulated in the presence of external dTMP as well (condition with MM like kinetics) (Figure 5B), which presumably suggests that even if allosteric modulation by these metabolites exists, it does not contribute to the observed cooperativity *in vivo*. We have included this information in the revised Discussion (Lines 421-425).

On the contrary, we feel that we are now in a position to put forward the metabolon idea with greater confidence mainly due to similar and different findings in other proteins (please refer to answer below to next question).

- It would help to have additional experiments (positive and negative controls, of sorts) that establish expectations for what the data in fig 7 should look like for enzymes known to be (or not) in metabolons. If you take an enzyme not thought to be in a metabolon, can you show that it obeys MM kinetics whether substrate is supplied exogenously (or not)? If you take an enzyme known to be in a metabolon, can you show that it behaves like Tmk - e.g. has the same Hill like dependence on substrate in the absence of external supplementation, and that this is lost in the presence of supplementation?

We thank the reviewer for this important suggestion. We have now included substrate vs product data for four other enzymes besides Tmk. These are Adenylate Kinase (AK), Uridylate Kinase (PyrH) and Adenylosuccinate Lyase (PurB) which show hyperbolic MM kinetics *in vivo* and GAR Transformylase (PurN) that shows highly cooperative Hill-like kinetics like Tmk. In addition, for AK, we show that supplementation by external AMP retains the hyperbolic behavior. These data are part of new Figure 7 now. While these additional data allow us to rule any artifact causing the Hill-like behavior for Tmk, it also allows us to put forth the metabolon idea much strongly due to the following reasons:

1. The mammalian analog of PurN protein is involved in the purinosome formation, which might explain the origin of the Hill-like kinetics of this protein. AK, on the other hand, is known to not be part of any pathway, and in our earlier pull-down (coIP-MS) experiments (Bhattacharyya et al, 2016, Elife), did not reveal any partner proteins, which presumably again justifies its hyperbolic *in vivo* kinetics.
2. Based on our previous work (Bhattacharyya et al, 2016, Elife), we also know that toxicity and filamentation upon over-expression is a hallmark of metabolon proteins. Based on some of our own data in this work and earlier (Bhattacharyya et al, 2016, Elife) as well as previous data from the ASKA libraries, we know that Tmk and PurN are both highly toxic upon over-expression and Tmk additionally leads to filamentation (Figure EV5). On the other hand, AK and PyrH are neutral to over-expression, while PurB is mildly toxic.

All the above data suggests that participation in a metabolon formation and subsequent diffusion limitation of one or more of substrate might shed light on the origin of the cooperative *in vivo* kinetics. We have included these points in the revised Discussion (Lines 461-479).

(3) The paper would benefit from an introduction that spends less time to on the broad problem of mapping genotype to phenotype but provides more context to the specific problem at hand. In particular:

- The authors cite heavily their own (nice) work on mapping biochemical parameters to phenotype, but should also mention other contributions to the rich literature on the topic, perhaps including work from Hartl (Dykhuizen et al (1987) Genetics v.115:25), and Anthony Dean (Lunzer et al (2005) Science v310:499)

We might have inadvertently missed out on some important references. We have included these and few other references along with appropriate discussion pertaining to genotype-phenotype relationship now.

- It would help to briefly review/provide context from the extensive prior literature on DHFR LOF, dTTP limitation, filamentation, and thymineless death

We have now included in the Introduction section a background of previous work concerning the effect of Trimethoprim on the *E. coli* metabolome and its effect on filamentation and TLD (Lines 59-68).

- The authors mention fractal kinetics but do not define/explain this term until much later

We have now referred to fractal kinetics only in the Discussion where we elaborate and explain it.

Minor concerns:

- dTMP vs TMP is at times easy to confuse. The authors might consider dTMP vs Tmp? Even changing capitalization might help readers to follow

We thank the reviewer for this helpful suggestion. Accordingly, we have renamed Tmp to denote Trimethoprim.

- It is sometimes feels inconsistent that the authors compare results for only some mutants and some temperatures and don't always include a WT control for each condition

We agree that for some experiments (particularly metabolomics and cell viability assay), we have not used the complete set of mutant strains, and instead used WT, W133V and I91L+W133V. This was mainly because in most of our assays, W133V and the double mutant V75H+I155A behave similarly (in terms of growth rates, extent of filamentation and ability to grow up to 42°C), while I91L+W133V and the triple mutant V75H+I91L+I155A behaved similarly. Hence, we chose one representative example from each of these two groups. Moreover, despite being a double mutant, strain I91L+W133V had the lowest growth rate among all mutants, which makes it an ideal candidate to study the extreme effect of mutations. Moreover, unlike W133V and V75H+I155A mutants which could grow up to 42°C, mutants I91L+W133V and V75H+I91L+I155A could only grow up to 40°C. We have clarified these points in the Methods section of the revised manuscript (Lines 788-790).

- The authors do not discuss the role of ATP in the reaction throughout the manuscript. While it seems clear from the mass spec data that ATP is not depleted relative to WT in the mutant cell lines, they should state this, and point out that their kinetics modeling assumes ATP is not limiting.

We thank the reviewer for pointing this out, and indeed this is an important assumption in our kinetic modeling. We have mentioned this in the revised manuscript (Lines 328-331).

- In several places they refer to the cells having DNA damage (e.g. lines 62-63, p. 3). While I agree this is very likely, it is not *shown*, so the authors should be a bit careful with their language.

It is true that we have explicitly shown that DNA in filamentous strains is damaged. Hence we now mention 'SOS response indicative of DNA damage' in the relevant places.

- On p.4/5 (section starting on Filamentous strains exhibit imbalance...) it would be helpful if the authors can explain their rationale for focusing/choosing two mutants out of those characterized in Fig. 1 and 2 for metabolomics

As we have mentioned in response to an earlier comment, W133V and I91L+W133V are representative examples from two groups of mutants. In addition, Figure 2 also includes

metabolomics data for WT treated with near-MIC concentration of Tmp, which mimics mutant strains in regard to filamentation.

- On page 5, lines 122-123, the parenthetical listing of mutants and conditions characterized is confusing. Could the authors just enumerate these with a numbered list?

We have now only mentioned strains and conditions in the figure (Figure 2) and have not reiterated them in the text.

- In Fig. 5 the authors use one type of assay (time dependent colony counts, panels B,C) to show loss of viability/TLD at high doses of TMP, but a different assay (dilution spots + outgrowth) to show a lack of TLD for DHFR LOF mutations. It would help to show the dilutions spots and outgrowth for the TLD condition (WT at high TMP, +/- GMP) as a reference/control in panel 5D. Panels 5B/C also really need titles.

Figure 4D (revised numbering) now also shows the dilution spots for WT treated with high (5µg/ml) Trimethoprim concentration, that shows loss in cfu in the presence of GMP, indicating TLD. Titles have also been added to Figure 4B/C.

- In Fig.2 - y axis needs units (presuming per minute as in Fig 1, but would be good to reiterate)

The units have been added now.

- P.9 line 248 - 20-fold overexpression, "overexpression" is not really the right word here, seems like over-abundance?

We have changed this to 20 fold higher levels.

- P. 11 line 306, say TMK is Hill-like under native conditions. I think the native conditions are unknown (or at least not quantitatively defined) and depend on the growth environment of E. coli (mammalian gut?) It is probably more accurate to say presence or absence of exogenous dTMP.

We have replaced 'native' with 'absence of external dTMP'.

Thank you for sending us your revised manuscript. We have now heard back from the three reviewers who were asked to evaluate your study. As you will see the reviewers are satisfied with the modifications made and think that the study is now suitable for publication.

Before we can formally accept your manuscript, we would ask you to address the following issues.

REFEREE REPORTS

Reviewer #1:

The points raised in the previous round of review have been satisfactorily addressed

Reviewer #2:

The authors have addressed my concerns

Reviewer #3:

I would like to congratulate the authors on a very rigorous and thorough revision. All of my concerns are now addressed, and I recommend this article for publication.

I particularly appreciate the addition of Fig. 7 which adds mass-spec derived in vivo activity curves for several other metabolic enzymes. It is nice to see confirmation that other enzymes do indeed behave in a Michaelis-Menten (hyperbolic) fashion as expected. The Hill-like graph for purN is quite striking. These data - taken alongside the evidence that the human ortholog is part of the purinosome - strengthen the authors proposal that Tmk is similarly part of a metabolon. I agree with their statement in the point-by-point response that "...we feel we are now in a position to put forward the metabolon idea with greater confidence mainly due to similar and different findings in other proteins."

I have only two minor suggestions remaining for the authors consideration:

1. The authors nicely explain their rationale for sometimes focusing on W133V and the double mutant I91L+W133V in the point-by-point response; however it would help general reader a lot to add a sentence or two describing this in the manuscript itself.
2. I think there is a typo on line 92 p. 4 - I believe the authors mean V75H+I155A rather than V75H+W133V.

The authors have made all requested editorial changes.

Accepted**11th May 2021**

Thank you again for sending us your revised manuscript. We are now satisfied with the modifications made and I am pleased to inform you that your paper has been accepted for publication.

Corresponding Author Name: EUGENE SHAKHNOVICH, SANCHARI BHATTACHARYYA

Journal Submitted to: MOLECULAR SYSTEMS BIOLOGY

Manuscript Number: MSB-2021-10200